# Isoferulic acid facilitates effective clearance of hypervirulent *Klebsiella pneumoniae* through targeting capsule

**Tingting Wang**[1,2☯], **Huaizhi Yang**[2☯], **Qiushuang Sheng**[3], **Ying Ding**[2], **Jian Zhang**[2], **Feng Chen**[2], **Jianfeng Wang**[2]*, **Lei Song**[1]*, **Xuming Deng**[1,2]*

**1** Department of Respiratory Medicine, Center for Pathogen Biology and Infectious Diseases, State Key Laboratory for Diagnosis and Treatment of Severe Zoonotic Infectious Diseases, The First Hospital of Jilin University, Changchun, China, **2** State Key Laboratory for Diagnosis and Treatment of Severe Zoonotic Infectious Diseases, Key Laboratory for Zoonosis Research of the Ministry of Education, Institute of Zoonosis, and College of Veterinary Medicine, Jilin University, Changchun, Jilin, China, **3** Jilin Province Product Quality Supervision and Inspection Institute, Changchun, Jilin, China

☯ These authors contributed equally to this work.

\* wjf927@jlu.edu.cn (JW); l.song@139.com (LS); Dengxm@jlu.edu.cn (XD)

**Data Availability Statement:** The raw RNA-seq data presented in this study have been deposited in the National Center for Biotechnology Information BioProject database and are available in the

## Abstract

Hypervirulent *Klebsiella pneumoniae* (hvKP) poses an alarming threat in clinical settings and global public health owing to its high pathogenicity, epidemic success and rapid development of drug resistance, especially the emergence of carbapenem-resistant lineages (CR-hvKP). With the decline of the "last resort" antibiotic class and the decreasing efficacy of first-line antibiotics, innovative alternative therapeutics are urgently needed. Capsule, an essential virulence determinant, is a major cause of the enhanced pathogenicity of hvKP and thus represents an attractive drug target to prevent the devastating clinical outcomes caused by hvKP infection. Here, we identified isoferulic acid (IFA), a natural phenolic acid compound widely present in traditional herbal medicines, as a potent broad-spectrum *K. pneumoniae* capsule inhibitor that suppresses capsule polysaccharide synthesis by increasing the energy status of bacteria. In this way, IFA remarkably reduced capsule thickness and impaired hypercapsule-associated hypermucoviscosity phenotype (HMV), thereby significantly sensitizing hvKP to complement-mediated bacterial killing and accelerating host cell adhesion and phagocytosis. Consequently, IFA facilitated effective bacterial clearance and thus remarkably protected mice from lethal hvKP infection, as evidenced by limited bacterial dissemination and a significant improvement in survival rate. In conclusion, this work promotes the development of a capsule-targeted alternative therapeutic strategy for the use of the promising candidate IFA as an intervention to curb hvKP infection, particularly drug-resistant cases.

## Author summary

Hypervirulent *Klebsiella pneumoniae* (hvKP) is becoming an imminent threat to public health because of the constant emergence of drug resistance and its devastating clinical

Sequence Read Archive (SRA) under BioProject accession number PRJNA1108905. Other data are available in the manuscript and supporting information.

**Funding:** This work was supported by grants from the National Natural Science Foundation of China 32402940 (to TW) and U22A20523 (to XD), the Interdisciplinary Integration and Innovation Project of Jilin University JLUXKJC2021QZ04 and Technology Development Project of Changchun City 23YQ10 (to LS). The funders had no role in study design, data collection and analysis, decision to publish, or preparation of the manuscript.

**Competing interests:** The authors have declared that no competing interests exist.

outcomes. Meanwhile, the both notorious phenotypes are integrated to create carbapenem-resistant and hypervirulent *K. pneumoniae* (CR-hvKP), which leads to a scarcity of effective treatments. Therefore, there is an increasing urgent need for new therapeutics other than antibiotics. The indispensable role of capsule in immune evasion and pathogenesis renders it a promising target for developing novel alternative strategies to combat hvKP infections. In this study, we identified the small natural compound isoferulic acid (IFA) as a potent broad-spectrum inhibitor of *K. pneumoniae* capsule. Hypercapsule-associated hypermucoviscosity (HMV), a characteristic phenotype that has been shown to play a distinct and critical role in the pathogenesis of hvKP, is significantly impaired by IFA. RNA-seq analysis revealed that IFA interferes with the biosynthesis of capsule polysaccharide through altering bacterial metabolism status. Consequently, IFA sensitizes hvKP to serum killing and impedes capsule- and HMV-mediated resistance to adherence and phagocytosis. In vivo, IFA treatment facilitates effective host clearance and thereby systemically protects mice against hvKP infection. Our research sheds light on the development of capsule-targeted drug and provides an effective lead compound that may contribute to the management of hvKP infections as an intervention.

## Introduction

*K. pneumoniae* is a ubiquitous and increasingly remarkable multidrug-resistant pathogenic bacterium capable of causing surgical site infection, pneumonia, bloodstream infection and other invasive infections. According to a recent analysis of data from the SENTRY Antimicrobial Surveillance Program over a 20-year period (1997–2016), *K. pneumoniae* ranks third behind *Escherichia coli* and *Staphylococcus aureus* as a major cause of bloodstream infections worldwide [1]. The extensive spread of *K. pneumoniae* is associated mainly with its ability to acquire new genetic elements, leading to the emergence of two typical pathotypes termed classical *K. pneumoniae* (cKP) and hypervirulent *K. pneumoniae* lineage (hvKP), each of which poses distinct clinical challenges. Although cKP is generally referred to as an opportunistic pathogen associated mainly with hospital-acquired infections in individuals who undergo major surgery or immunosuppression, the frequent emergence and rapid spread of new antimicrobial resistance greatly improves its clinical significance. In particularly, carbapenem resistant cKP (CRKP) has topped the list of global urgent antibiotic-resistant threats [2]. In contrast, hvKP is much more virulent than cKP, with the ability to cause several invasive infections both in hospital patients and otherwise healthy individuals in community settings. Clinically, hvKP is notorious for causing pyogenic liver abscesses and disseminating to the blood, brain, lungs, eyes and other organs, with a high mortality rate ranging from 3% ~ 55% and severe complications, even resulting in several severe morbidities in survivors, which is uncommon for most gram-negative pathogens [3]. Therefore, the hypervirulent lineage raises another critical clinical challenging risk because of its high pathogenicity, and it has been classified as a predominant pathogen.

HvKP has spread globally since it was originally reported in the 1980s in Taiwan, but it is most prevalent in the Asian Pacific Rim, especially in China [4]. Reportedly, the prevalence of hvKp among *K. pneumoniae* infections is gradually increasing, ranging from 12 to 45% in endemic areas [5], which poses a devastating clinical challenge. Although the evolving pathotype was initially sensitive to antibiotics, emerging evidence suggests that hvKP is becoming antibiotic resistant because of the acquisition of resistance plasmids or genetic insertion of resistance elements. Alarmingly, the two concerning pathotypes of carbapenem resistance and

hypervirulence are gradually converging, yielding another devastating *K. pneumoniae* lineage, carbapenem-resistant hvKP (CR-hvKP), which causes several serious clinical concerns. Currently, novel alternative therapeutic strategies are urgently needed to curb hvKP infection, especially in antibiotic-resistant cases.

Unlike other clinically critical pathogens, such as *Staphylococcus aureus* or *Salmonella typhimurium*, which employ plenty of extensively studied virulence factors to establish infection in the host, *K. pneumoniae* mainly utilizes four major classes of virulence determinants for adhesion, invasion and protection from host clearance, including capsule, lipopolysaccharide (LPS), fimbria and siderophores [6]. Among these virulence factors, the capsule is also known as the K-antigen and is termed the basis for typing serotypes according to the strain-specific composition of capsule polysaccharide. The capsule of *K. pneumoniae* is synthesized via a Wzx/Wzy-dependent pathway encoded by the *cps* gene cluster located on the chromosomal operon, which exhibits genetic difference between distinct serotypes but has similar biosynthesis and export machinery [7]. To date, more than 80 serotypes have been identified using K-antigen-based serological methods, among which the K1 and K2 strains are generally more virulent than others and are thus the most prevalent clinical isolates in hvKP infections [8]. Typical features associated with hvKP include abundant capsule production, also termed hypercapsule, hypermucoviscosity and increased variety and number of siderophores. It was clearly established that the increase in capsule production was responsible for the enhanced pathogenicity of hvKP [9]. Enclosing the outermost layer of *K. pneumoniae* in the form of a polysaccharide matrix, capsule protects bacterial cells against extreme environments and host immune clearance as a protective barrier and virulence. First, capsule mediates immune evasion through blocking phagocytosis by immune cells owing to its electronegativity, preventing the bactericidal activity of complement system and host-derived antimicrobial peptides, enhancing resistance to the killing of reactive oxygen species (ROS) and mimicking host glycans as an antigen to subvert host anti-infection responses [10–12]. More importantly, recent advances in hvKP pathogenesis revealed that capsule can also actively impede host immune responses through multiple mechanisms, including dampening the activation of NF-κB-controlled inflammation response[13], manipulating phagosome fusion with lysosomes [12], subverting host immune defenses through the suppression of the SUMOylation of host proteins [14], and last but not the least, skewing the polarization of lung macrophages to promote infection [15]. These pathogenicity mechanisms of capsule facilitate bacterial dissemination during hvKP infection and contribute to the development of drug resistance. The multifaceted critical role of capsule in *K. pneumoniae* pathogenesis renders it an attractive therapeutic target for combating hvKP infection.

Isoferulic acid (3-hydroxy-4-methoxycinnamic acid, IFA) is a rare kind of phenolic acid identified as the main active ingredient of the rhizome of *Cimicifuga heracleifolia*, an herbal medicine frequently used in traditional medicine in oriental countries such as China and Japan [16]. Previous studies revealed that the natural small molecule compound possess antiviral [17], anti-hematologic malignancy [18], anti-glycation and antioxidation properties [19,20]; however, there is no research on the therapeutic potential of IFA for the treatment of bacterial infections. In this study, we performed high-throughput biochemical screening and identified IFA as a potent broad-spectrum inhibitor of *K. pneumoniae* capsule that reduces capsular polysaccharide production through increasing the energy status of *K. pneumoniae*. Accordingly, IFA treatment significantly improved host immune clearance against hvKP infection and therefore effectively protected mice from lethal hvKP infection. We propose that the newly identified inhibitory effect of IFA on *K. pneumoniae* capsule synthesis may provide a novel therapeutic option to curb *K. pneumoniae* infections, which is particularly essential for antibiotic-resistant hvKP and CRKP cases.

## Results

### IFA effectively decreased the production of *K. pneumoniae* capsule

The indispensable role of capsule in the pathogenicity of *K. pneumoniae* renders it an attractive drug target, yet no effective capsule inhibitor has been identified thus far. Herein, we screened potential anti-CPS inhibitors using a uronic acid assay and identified the natural compound IFA as the most promising one (Figs 1A and S1), which dose-dependently reduced the amount of CPS in the hypervirulent *K. pneumoniae* strain K7 (Fig 1B), but did not exhibit any visible inhibitory effect on bacterial growth and had an MIC exceeding 1024 μg/ml (Fig 1C). Consistently, SDS–PAGE analysis confirmed that the amount of polysaccharide was significantly decreased by IFA treatment (Fig 1D). To profile the changes in capsule architecture, we observed capsule transmission using electron microscopy (TEM). As shown in Fig 1E, wild-

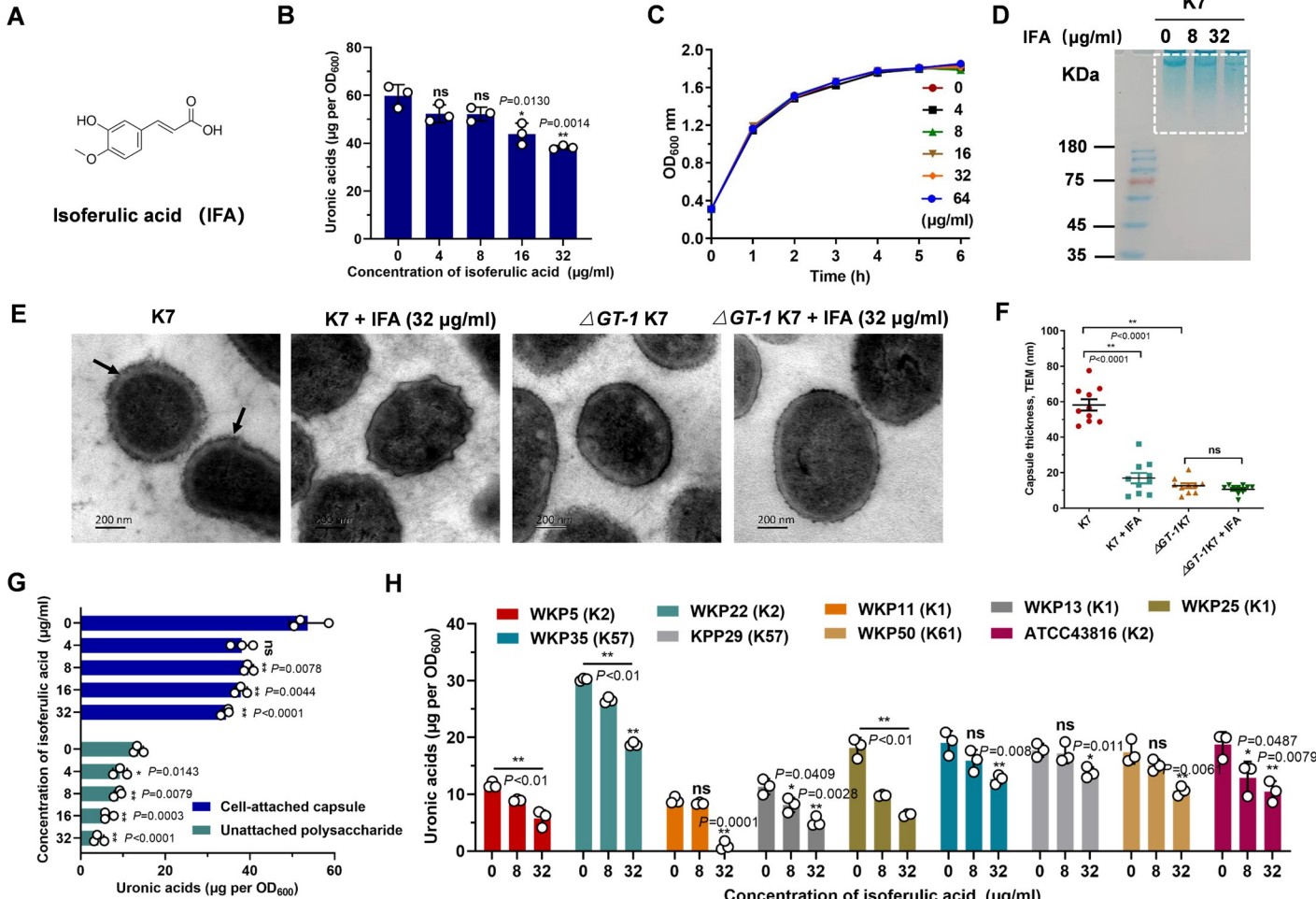

**Fig 1. IFA effectively reduced the production of *K. pneumoniae* capsule.** (A) Chemical structure of IFA (IFA). (B) Uronic acid assay to confirm the capsule production of *K. pneumoniae* K7 with the treatment of indicated concentrations of IFA. (C) Growth curves of *K. pneumoniae* K7 in the presence of the indicated concentrations of IFA. (D) Alcian blue staining of capsule samples extracted from *K. pneumoniae* K7 with or without IFA treatment. (E) Transmission electron microscopy images of wild-type *K. pneumoniae* K7 and *ΔGT-1* K7 in the presence of DMSO or 32 μg/ml IFA. The black arrows indicate capsule, and its thickness was measured using ImageJ (F). (G) Comparison of cell-attached and unattached capsule content of *K. pneumoniae* K7 with or without IFA treatment via a uronic acid assay. (H) Capsule production of different *K. pneumoniae* strains with or without IFA treatment. The data are presented as the means ± SEMs. One-way ANOVA and Tukey's posttest was carried out to determine the statistical significance of different groups. *$P < 0.05$, **$P < 0.01$ and ns, no significance compared with the DMSO control group.

type K7 *K. pneumoniae* was encased in a thick and filamentous capsule, while a dramatic reduction in capsule thickness was observed in cells treated with IFA (Fig 1F), which exhibited greater similarity to the capsule-deficient mutant with deletion of the GT-1 glycosyltransferase (*ΔGT-1* K7), with barely visible capsule boundaries around the cells.

To explore whether IFA affects the process of capsule synthesis or retention, we measured cell-attached and unattached CPS, respectively. As with the changes in total CPS (Fig 1B and 1D), both the amounts of capsule bound to the cell and free in the supernatant were dose-dependently suppressed by IFA, indicating that IFA might inhibit capsule synthesis but not the retention process (Fig 1G). To test whether the inhibitory effect of IFA on capsule synthesis was serotype specific, we further performed a uronic acid assay on other strains of different serotypes, including K1, K2, K57 and K61. Promisingly, the amounts of CPS in all the tested strains were dose-dependently reduced by IFA (Fig 1H), suggesting that IFA is a potent broad-spectrum inhibitor of *K. pneumoniae* capsule. In conclusion, these results confirmed that IFA is an effective and serotype-independent capsule inhibitor that significantly reduced capsule thickness by inhibiting CPS production.

## IFA impaired hypercapsule-associated mucoviscosity phenotypes of *K. pneumoniae* via capsule inhibition

The hypermucoviscosity phenotype (HMV) is a typical feature of hvKP characterized by poor recovery and sedimentation when centrifuged, and this phenotype has been shown to play a distinct role in *K. pneumoniae* pathogenesis. Although several recent studies have demonstrated discordant changes in capsule production and HMV and also found some new potential regulatory mechanisms [21,22], a genome-wide genetic screening revealed that capsule biosynthesis and HMV were coordinated in most cases [23]. To explore whether IFA also affected the HMV phenotype of *K. pneumoniae*, we further performed sedimentation assay. Consistent with the changes in capsule production, centrifugation resistance, referred to here as an indicator of excessive mucoidity, was also significantly reduced by increasing dose of IFA (Fig 2A), and this suppressive effect on HMV was also confirmed in different serotypes (Fig 2B).

Building upon the above findings, we next performed a string test to further confirm the inhibitory effect of IFA on HMV. Consistent with previous reports, wild-type K7 formed obviously more large and mucoid colonies on blood agar plates, and these colonies could be easily stretched to an average length of more than 5 mm, which is commonly regarded as a criterion for HMV (Fig 2C). In addition, the wild-type K7 colonies were large, protuberant and more mucoid on the LB agar plates, whereas the colonies in IFA group were smaller, flat, grayish and translucent, similarly to those in the *ΔGT-1* K7 group (Fig 2D). Interestingly, the surface of the wild-type K7 bacteria was covered by a mucoid matrix, thus, the cell boundary was not clear according to SEM (Fig 2E). In contrast, the morphology and structure of the bacteria were not impacted by 32 μg/ml IFA, but the mucoid matrix masking the bacterial surface was prominently eliminated, with a clear boundary, as in *ΔGT-1* K7.

Given that HMV not only depends on hypercapsule production but also requires the small protein regulator RmpD, we constructed a *rmpD*-deficient mutant to exclude the possibility that IFA suppresses HMV via acting on rmpD rather than capsule. Consistent with the findings of previous study, *ΔrmpD* K7 *K. pneumoniae* completely lost HMV phenotype (S2A–S2C Fig). Interestingly, IFA treatment still reduced uronic acid level of *ΔrmpD* K7 in a dose-dependent manner, suggesting that IFA suppressed HMV by affecting capsule (S2D Fig). Together, these results elucidated that IFA treatment effectively impaired HMV via reducing CPS production.

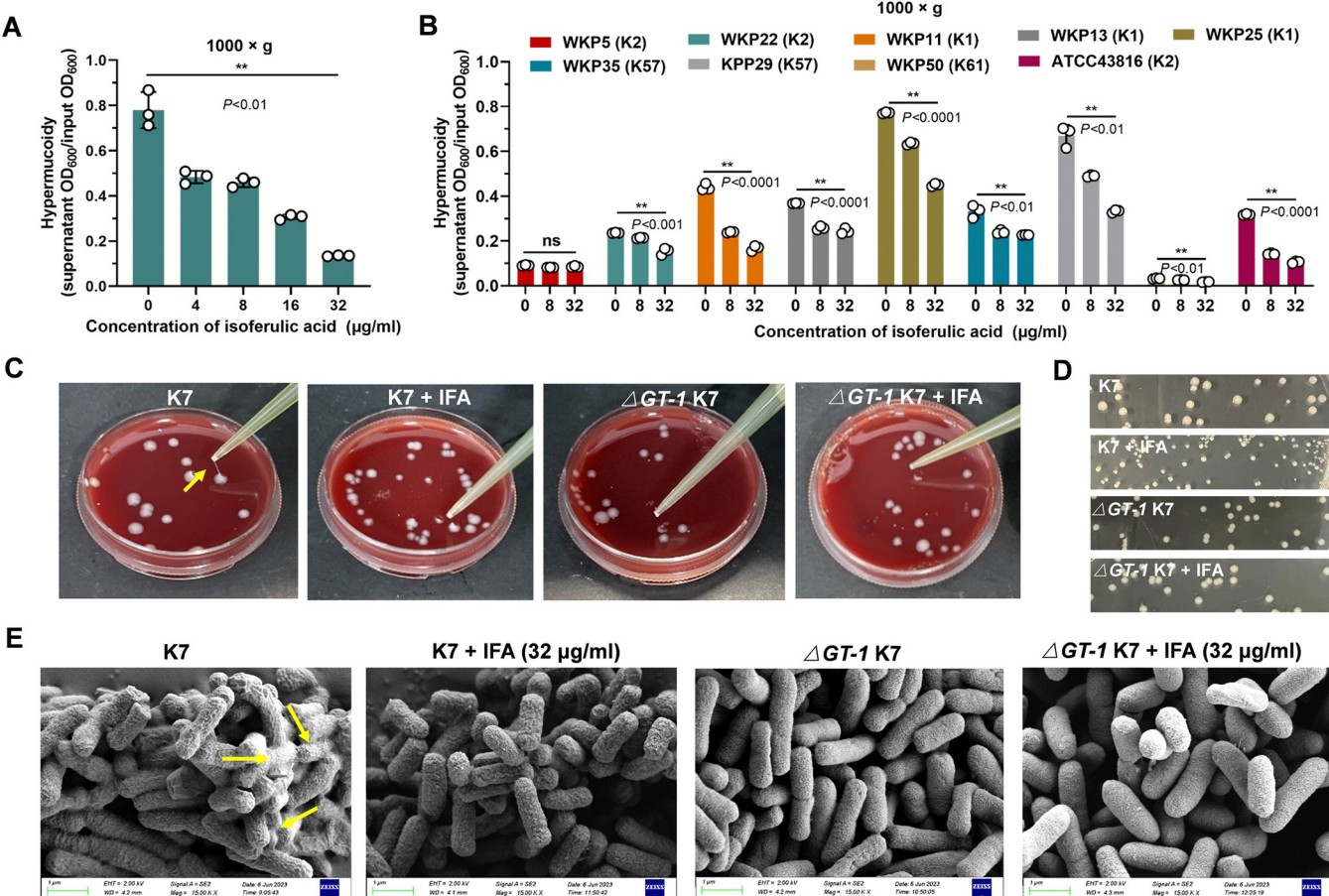

**Fig 2. IFA abrogated capsule-associated hypermucoviscosity phenotypes of hypervirulent *K. pneumoniae*.** (A) Hypermucoidy assay of hypervirulent *K. pneumoniae* K7 treated with DMSO or indicated concentrations of IFA. Bacterial cultures were centrifuged at 1,000 × g in a fixed angle rotor for 5 min to measure the $OD_{600}$ of the supernatants, which was defined relative to the initial $OD_{600}$. (B) Hypermucoidy analysis of different serotypes of *K. pneumoniae*. (C) Representative colony phenotypes of *K. pneumoniae* K7 and *ΔGT-1* K7 on blood agar plates containing DMSO or 32 μg/ml IFA; the "string test" was performed at the same time. The yellow arrow indicates the string stretched by the tips. (D) The colony morphologies of *K. pneumoniae* K7 and *ΔGT-1* K7 on LB agar plates containing DMSO or 32 μg/ml IFA are shown. (E) Scanning electron microscopy images of wild-type *K. pneumoniae* K7 and *ΔGT-1* K7 in the presence of DMSO or 32 μg/ml IFA. The data are presented as the means ± SEMs. One-way ANOVA and Tukey's posttest was performed to determine the statistical significance of different groups. **$P < 0.01$ and ns, no significance compared with the DMSO control group.

## IFA reduced CPS biosynthesis through altering bacterial metabolism

To explore how IFA reduced the thickness of *K. pneumoniae* capsule and capsule-associated HMV, we further analyzed the transcriptome of wild-type K7 *K. pneumoniae* after different treatments. Accordingly, 295 differentially expressed genes (DEGs) were identified between DMSO- and IFA (32 μg/ml)-treated bacteria, comprising 78 upregulated and 217 downregulated genes in the presence of IFA (Fig 3A). The heat cluster of these DEGs further revealed significant transcriptome alterations caused by IFA (Fig 3B). Gene Ontology (GO) enrichment analysis of these DEGs revealed the top 20 GO terms, including the metabolic and catabolic processes of organic substances and subsequent oxidation–reduction processes (Fig 3C), among which the oxidation–reduction process was the most affected, with 8 upregulated and 23 downregulated genes (adjusted $p$ = 5.73E-05). Cellular metabolism and oxidation–reduction processes are closely related to energy status, and reportedly, high intracellular ATP level is a limiting factor for CPS production and HMV [24]. Hence, we further evaluated the

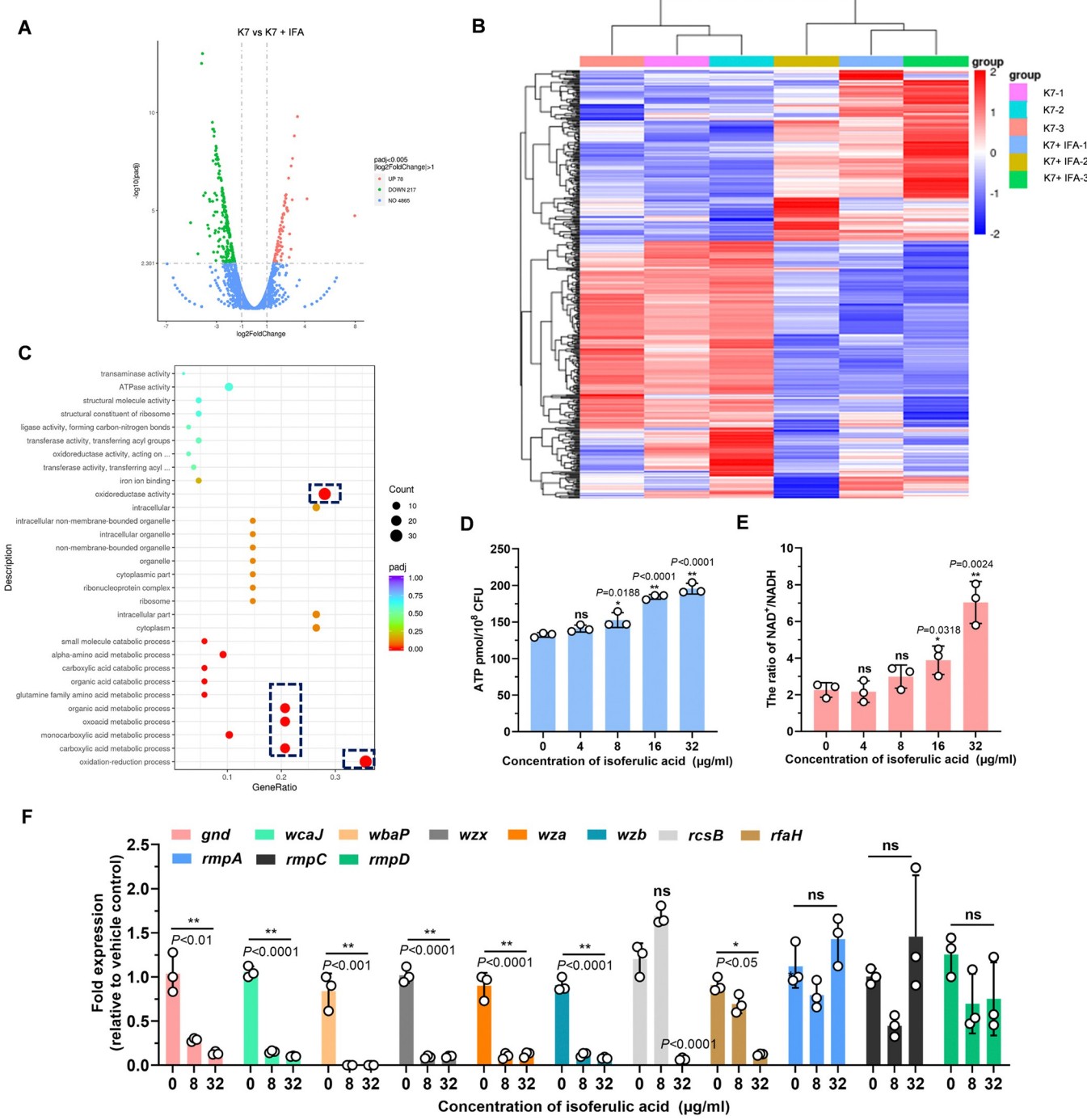

**Fig 3. IFA reduced the capsular biosynthesis through altering bacterial metabolism.** (A) Volcano plot analysis of differentially expressed genes (DEGs) between DMSO- or 32 μg/ml IFA-treated *K. pneumoniae* K7. A total of 295 differentially expressed genes (DEGs), including 78 upregulated and 217 downregulated genes, were identified in the IFA-treated group. (B) Heat cluster of DEGs. (C) The top 20 enriched Gene Ontology (GO) terms of all DEGs compared to DMSO control. (D) Intracellular ATP levels in *K. pneumoniae* K7 cells treated with DMSO or indicated concentrations of IFA. (E) The ratio of NAD$^+$/NADH in *K. pneumoniae* K7 treated with DMSO or indicated concentrations of IFA. (F) The transcriptional levels of capsule synthesis genes and rmp genes. The data are presented as the means ± SEMs. One-way ANOVA and Tukey's posttest was performed to determine the statistical significance of different groups.*$P < 0.05$, **$P < 0.01$ and ns, no significance compared with DMSO control group.

intracellular ATP levels of log-phase *K. pneumoniae*. As expected, the ATP level of *K. pneumoniae* was predominantly increased by IFA in a dose-dependent manner (Fig 3D), and in accordance with this result, the ratio of NAD$^+$/NADH was also greater in IFA-treated bacteria (Fig 3E). These results further again uncovered the internal integration between CPS biosynthesis and cellular metabolism and energy status.

Next, we detected the transcriptional levels of core capsule synthesis genes and rmp genes. The qRT-PCR data demonstrated that the transcriptional levels of *gnd*, *wcaJ*, *wbaP*, *wzx*, *wza*, *wzb*, *rcsB* and *rfaH* were dose-dependently reduced by IFA, whereas the levels of *rmpA*, *rmpC* and *rmpD* were not affected (Fig 3F), indicating that IFA down-regulated capsule synthesis genes but not HMV-related genes. Taken together, these results suggested that IFA reduced CPS biosynthesis and HMV through suppressing the transcriptional levels of capsule cluster genes, which could be attributed to the alteration in bacterial metabolism and energy status.

## IFA sensitized *K. pneumoniae* to serum killing via suppressing capsule

The complement system present in serum and tissue fluids is the first line of host defense against bacterial invasion. Nevertheless, several gram-negative bacteria have evolved multiple strategies to resist complement-mediated killing through interfering with the activation of the complement cascade or degrading already activated complement molecules [25], which contributes to the survival, multiplication and dissemination of bacteria. Capsule is a critical determinant for *K. pneumoniae* survival in serum, conferring strong resistance to the binding of C3b and C5b-9 to the cell surface [26]. The remarkable suppression of IFA on capsule synthesis prompted us to further investigate whether IFA sensitized *K. pneumoniae* to serum killing. Consistently, the viability of the *ΔGT-1* K7 strain, which is deficient in capsule production, was significantly reduced in 20% normal human serum (NHS) within 60 min and was eliminated at 120 min, while the serum killing of wild-type K7 was obviously delayed compared to the capsule-deficient mutant and a sharp decrease was observed until 120 min (Fig 4A). As expected, IFA treatment dose-dependently decreased the resistance of wild-type K7 to serum killing. Notably, IFA (64 µg/ml) did not have an obvious effect on the growth of *K. pneumoniae* either in LB medium or heat-inactivated NHS (56˚C, 30 min) (Fig 4B), again supporting its minimal antimicrobial activity.

All three known complement pathways converge at C3 cleavage, with eliciting C3b generation and covalent binding to the target bacterial surface, followed by the assembly of C5 convertases and the appearance of a C5b cleavage product that subsequently forms a C5b-9 membrane attack complex capable of lysing cells [27]. Hence, we examined the deposition of the C3b/C3bi and C5b-9 complexes by immunofluorescence microscopy analysis. Consistent with the serum killing results, serum exposure of *ΔGT-1* K7 exhibited considerable binding of C3b/C3bi and C5b-9 within the first hour, while the wild-type K7 strain displayed little to no binding at this time point and only slightly increased over time (Fig 4C and 4D). Promisingly, treatment with 32 µg/ml IFA led to a significant increase in the amount of complement binding to wild-type K7 but not to *ΔGT-1* K7 (Figs 4C and 4D and S3). Similarly, these observations were further validated by flow cytometry, which yielded the same results (Fig 4E–4H). Collectively, these results demonstrated that IFA increased the recruitment of complement molecules to the *K. pneumoniae* surface and thus sensitized bacteria to complement-mediated killing by inhibiting CPS.

## IFA impeded capsule-mediated immune evasion of *K. pneumoniae*

Capsule- and HMV-mediated blockade of adherence and phagocytosis is one of the most important pathogenesis mechanisms of hvKP [22]. To determine whether IFA disarms this in

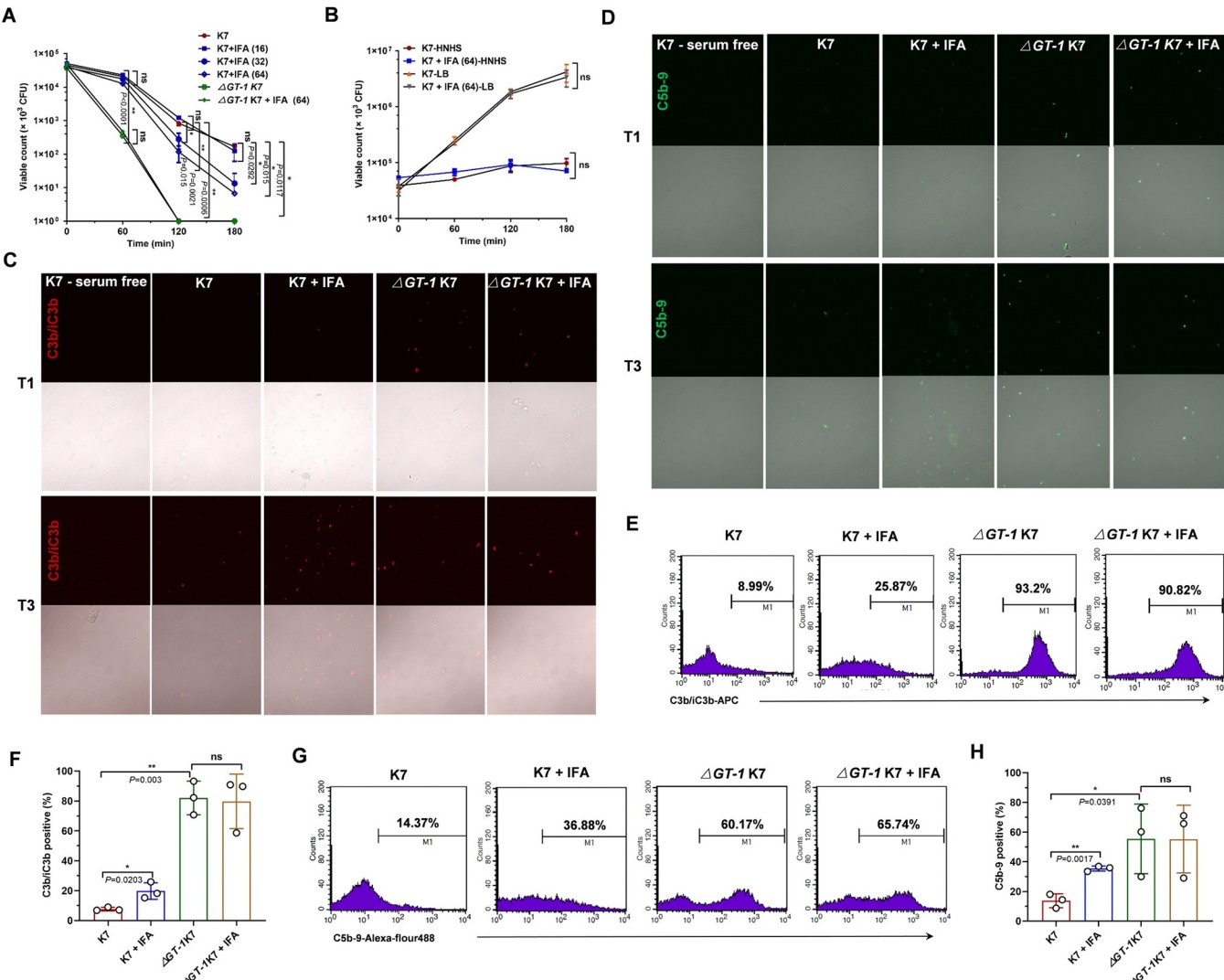

**Fig 4. IFA sensitized hypervirulent _K. pneumoniae_ to complement-mediated killing by inhibiting capsule.** (A) Serum-killing assay of _K. pneumoniae_ K7 and _ΔGT-1_ K7 in the presence of DMSO or indicated concentrations of IFA. Bacteria were incubated with 20% NHS, and the viable count was determined by serial dilution and microbiological plating at the indicated time points. (B) Bacterial viability of _K. pneumoniae_ K7 in heat-inactivated NHS (HNHS) and LB medium supplemented with DMSO or 64 μg/ml IFA. (C) Immunofluorescence microscopy analysis of C3b/C3bi deposition on the surface of 1-hour and 3-hour serum-exposed _K. pneumoniae_ K7 and _ΔGT-1_ K7 with the treatment of DMSO or 32 μg/ml IFA. C3b/C3bi was stained with APC-conjugated anti-C3b/ C3bi antibody. (D) Immunofluorescence microscopy analysis of C5b-9 formation on the surface of 1-hour- and 3-hour serum-exposed _K. pneumoniae_ K7 and _ΔGT-1_ K7 with the treatment of DMSO or 32 μg/ml IFA. The C5b-9 complex was stained with a mouse anti-C5b-9 antibody and Alexa Fluor 488-conjugated goat anti-mouse IgG. The fluorescence images (top) and transillumination images (bottom) were normalized within each panel. (E) Flow cytometry-based determination of C3b/C3bi binding to _K. pneumoniae_ K7 and _ΔGT-1_ K7 with the treatment of DMSO or 32 μg/ml IFA was performed after a 3-hour incubation in 20% NHS at 37˚C. (F) The data were analyzed using unpaired two-tailed Student's t-test. (n = 3). (G) Flow cytometry-based determination of C5b-9 formation on _K. pneumoniae_ K7 and _ΔGT-1_ K7 with the treatment of DMSO or 32 μg/ml IFA was performed after 3 hours of incubation in 20% NHS at 37˚C. (H) The data were analyzed using unpaired two-tailed Student's t-test (n = 3). The means ± SEMs are shown. *$P < 0.05$, **$P < 0.01$ and ns, no significance.

vitro virulence phenotype, we performed an adherence assay using human A549 alveolar epithelial cells and performed phagocytosis assays with mouse J774 macrophages and mouse primary peritoneal macrophages (MPMs), respectively. As shown in Fig 5A, more _ΔGT-1_ K7 mutant attached to host cells compared with wild-type K7 strain, while IFA treatment significantly increased the adherence of wild-type K7 but not _ΔGT_-1 K7. In addition, fewer than 10%

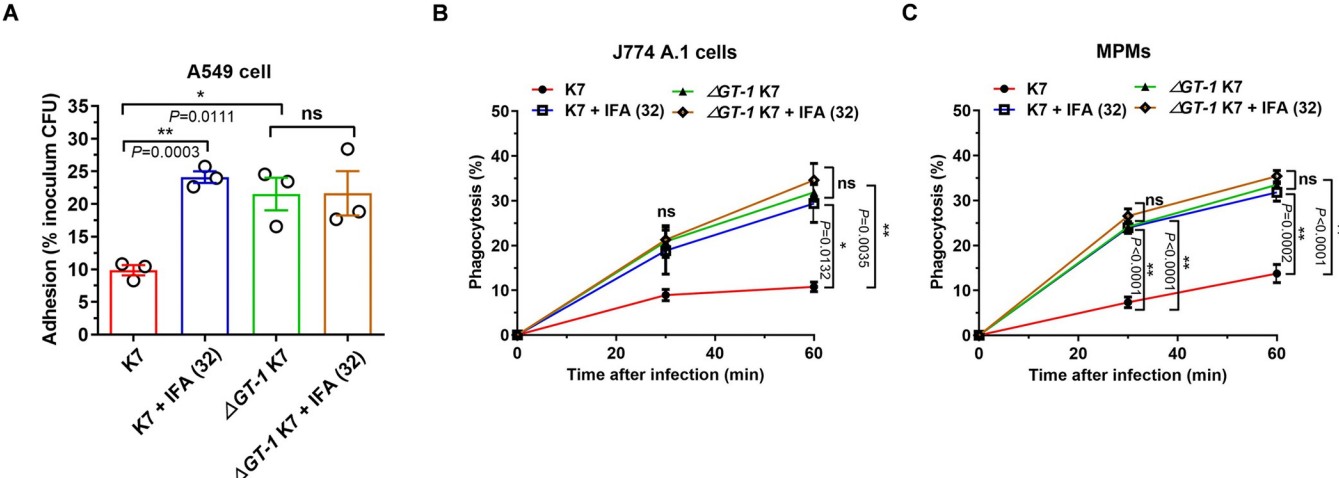

**Fig 5. IFA disarmed capsule-mediated immune evasion.** (A) Adhesion assay of *K. pneumoniae* K7 and *ΔGT-1* K7 to A549 cells in the presence of DMSO or 32 μg/ml IFA. Cells were infected with *K. pneumoniae* K7 or *ΔGT-1* K7 at an MOI of 50 for 2 hours and then washed, lysed and plated on LB agar plates to quantify the number of colony-forming units. Data are presented as a percentage of the initial inoculum CFU. (B) Phagocytosis of *K. pneumoniae* K7 and *ΔGT-1* K7 by J774 macrophages in the presence of DMSO or 32 μg/ml IFA. The cells were infected at an MOI of 5 for 2 hours and then washed, followed by a further incubation in medium containing gentamicin (100 μg/ml) to kill the extracellular bacteria. The cells were then rinsed, lysed and plated on LB agar plates after serial dilution. (C) Phagocytosis of *K. pneumoniae* K7 and *ΔGT-1* K7 by mouse primary peritoneal macrophages (MPMs) in the presence of DMSO or 32 μg/ml IFA was also analyzed as described above. The data are presented as the means ± SEMs. Unpaired two-tailed Student's t-test was performed to determine the statistical significance of two groups. *$P < 0.05$, **$P < 0.01$ and ns, no significance.

of the wild-type K7 were phagocytized by J774 cells at 60 min post infection, whereas about 30% *ΔGT-1* K7 were intracellular at the same time (Fig 5B), confirming the effective blockade of phagocytosis by capsule. Surprisingly, IFA treatment increased the phagocytosis ratio to a level approaching that of the acapsular mutant, indicating the effective abrogation of capsule and its related HMV. Interestingly, a significant difference between wild-type K7 and *ΔGT-1* K7 appeared at 30 min post infection in MPMs cells, and IFA treatment also caused stark increase in wild-type K7 (Fig 5C).

Although the earlier study raised that HMV might be the main factor blocking host cell adherence during hvKP infection [22], our results demonstrated IFA still dose-dependently increased the adherence level of *ΔrmpD* K7 (S2E Fig). Of note, the deletion of *rmpD* indeed resulted in a significant increase in the amount of attached bacterial versus wild-type K7 (S2E Fig). Similarly, more *ΔrmpD* K7 were engulfed by J774 A.1 macrophage, and the phagocytosis ratio was also significantly increased by IFA (S2F Fig), indicating that IFA-mediated enhancement of adherence and phagocytosis might mainly depend on capsule rather than HMV, which requires both capsule and rmpD.

Collectively, these in vitro results confirmed that IFA disarms capsule-mediated prevention of adherence and phagocytosis, which would significantly impede the immune evasion of hvKP.

## In vivo IFA therapy rescued *G. mellonella* from *K. pneumoniae* infection

Based on the above in vitro virulence-associated analysis, we then performed a *G. mellonella* killing assay to evaluate the in vivo effect of IFA. *G. mellonella* were treated with 50 mg/kg IFA in 10 μl of vehicle or an equal volume of vehicle immediately after challenge with $2 \times 10^4$ wild-type K7 or *ΔGT-1* K7 bacteria. Wild-type K7 infection caused 100% mortality within 32 hours, whereas all the *ΔGT-1* K7-infected *G. mellonella* survived, which confirmed the great importance of capsule for full virulence of *K. pneumoniae* (Fig 6A). Promisingly, IFA treatment

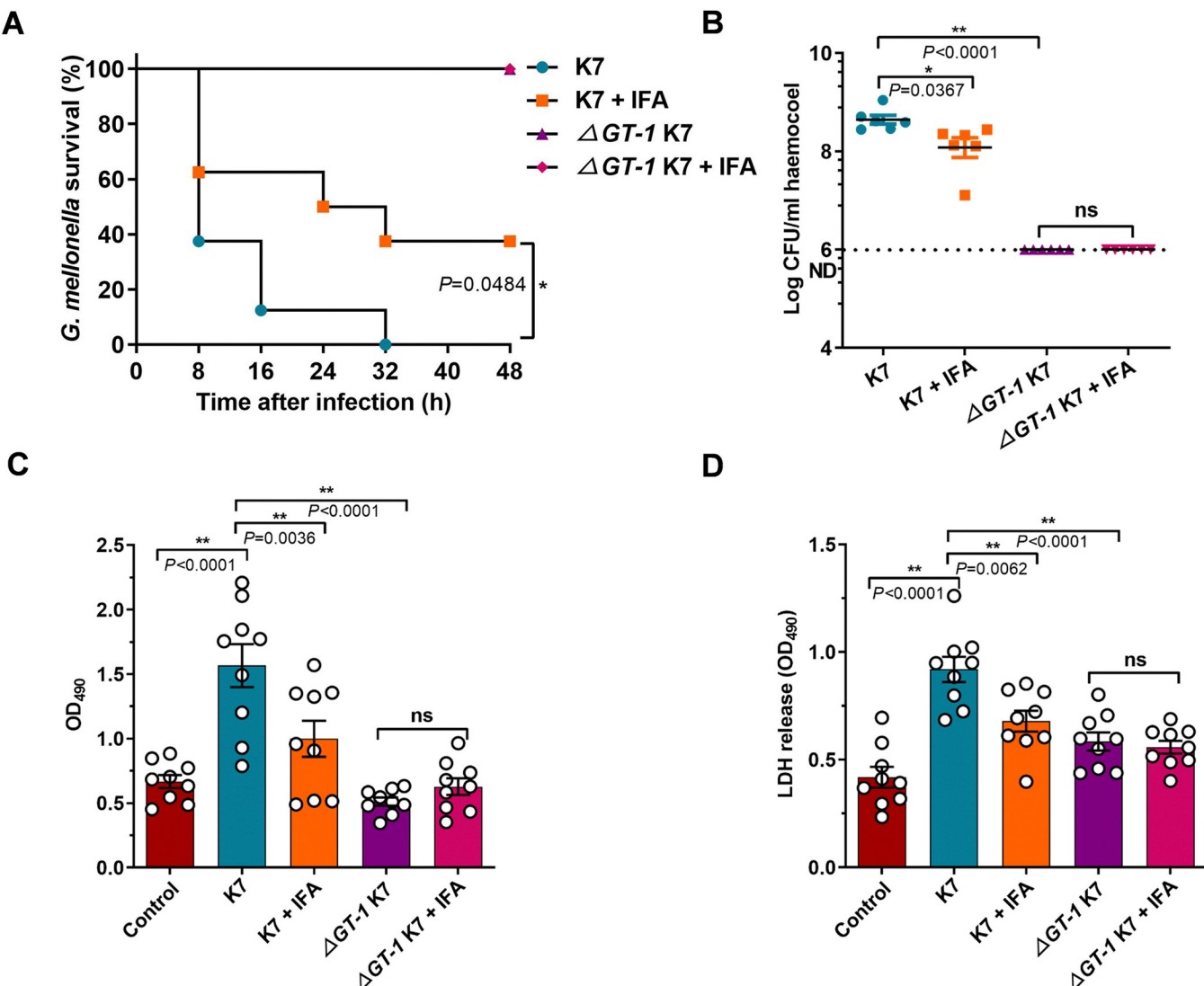

**Fig 6. IFA reduced the virulence of hypervirulent *K. pneumoniae* in *G. mellonella* larvae.** (A) Survival of *G. mellonella* 48 h after *K. pneumoniae* infection. *G. mellonella* were treated with 50 mg/kg IFA in 10 μl of 10% DMSO containing vehicle (10% DMSO, 45% stroke-physiological saline solution, 40% PEG400 and 5% Tween-80) or an equal volume of vehicle immediately after challenge with $2 \times 10^4$ wild-type *K. pneumoniae* K7 or *ΔGT-1* K7 bacteria, and the number of deaths was recorded for survival analysis. Statistical analysis was performed using the log-rank (Mantel–Cox) test (n = 8 larvae each group). (B) Bacterial load in the hemocoels of larvae (presented as cfu/ml). *G. mellonella* infected and treated as described above were sacrificed at 5 hours postinfection, and the hemocoels were serially diluted and microbiologically plated (n = 6 larvae each group). ND, not detected. The back lines presents the means ± SEMs. (C) Infection-induced melanization of hemocoels from the indicated groups. *G. mellonella* infected and treated as described above were sacrificed at 5 hours postinfection, and the hemocoels were collected to measure the degree of melanization at $OD_{490nm}$ (n = 9 larvae each group). The larvae in control group were injected with PBS and treated by equal volume of vehicle. (D) LDH release in the hemocoel from the indicated groups. *G. mellonella* infected as described above were treated with 50 mg/kg IFA in 10 μl of 1% DMSO containing vehicle (1% DMSO, 54% stroke-physiological saline solution, 40% PEG400 and 5% Tween-80) or an equal volume of vehicle and were sacrificed at 5 hours postinfection. Subsequently, the hemocoel was collected to measure LDH release using a cytotoxicity detection kit (n = 9 larvae each group). The results are presented as the $OD_{490nm}$ values. The data are presented as the means ± SEMs. One-way ANOVA and Tukey's posttest was used to perform multiple comparisons. *$P < 0.05$, **$P < 0.01$ and ns, no significance.

effectively protected *G. mellonella* from *K. pneumoniae* killing, with a 37.5% increase in the survival rate (Fig 6A). In addition, the bacterial burden in the hemocoel was determined at 5 hours post infection when all the *G. mellonella* were alive. As shown in Fig 6B, the number of wild-type K7 bacteria was much greater than the input number, but IFA treatment significantly reduced bacterial dissemination. In contrast, the number of *ΔGT-1* K7 bacteria was

lower than the input number and was not influenced by IFA, indicating strong host clearance against bacteria deficient in capsule. Melanization is a widely used hallmark reaction of *G. mellonella* for the assessment of bacterial profile and immune evasion, and is positively correlated with pathogenicity [28]. Consistent with the result of bacterial loading assay, wild-type K7 infection elicited the highest degree of melanization (Fig 6C). In contrast, and as expected, *ΔGT-1* K7-induced melanization was much lower compared with wild-type K7, and K7-caused melanization was also significantly attenuated by IFA treatment (Fig 6C). Lactate dehydrogenase (LDH) release, a marker of cell death, is often referred to as another predictor of virulence. We found that, compared with *ΔGT-1* K7, wild-type K7 elicited more LDH release, but this effect was significantly diminished by IFA treatment (Fig 6D). Together, these results suggested that IFA treatment effectively rescued *G. mellonella* from *K. pneumoniae* killing via inhibiting capsule-conferred pathogenicity.

## IFA conferred significant protection against *K. pneumoniae* infection in mice

Next, we further used mice lethal *K. pneumoniae* pneumonia model to better confirm the in vivo therapeutic effect of IFA. Similarly, challenge with $1 \times 10^7$ wild-type K7 killed all the mice treated with vehicle within 72 hours, but all the mice infected with *ΔGT-1* K7 survived. Promisingly, the survival rate of K7-infected mice was effectively enhanced by IFA treatment (50 mg/kg), with a significant increase of 41.67% (Fig 7A). The bacterial loads were subsequently detected with a 1/4 lethal dose of *K. pneumoniae* 40 hours post infection when all the mice were alive. Consistent with the 100% survival, almost no bacterial colonization was detected in the lungs of mice infected with *ΔGT-1* K7 (Fig 7B), and as expected, IFA treatment reduced the high bacterial burden of wild-type K7 approximately by 22 times. Moreover, we also detected the number of bacteria in the BALF, livers and spleens from different groups. Consistently, considerable bacterial loads were detected in the BALF and distal organs of the mice infected with wild-type K7 (Fig 7C–7E), while *ΔGT-1* K7 showed very little to no dissemination from lung to other organs, which indicated effective host clearance due to capsule deficiency. In contrast and as anticipated, IFA treatment significantly reduced the bacterial load in the BALF, livers and spleens of the mice challenged with wild-type K7 (Fig 7C–7E). Pathology analysis of lung sections from the mice infected with wild-type K7 exhibited more severe bronchopneumonia compared to *ΔGT-1* K7-infected mice (Fig 7F), accompanied by higher levels of airway inflammation (Fig 7G), neutrophil infiltration (Fig 7H), intralesional bacterial burden and, as a result, a more severe overall histopathology score (Fig 7I and 7J). Collectively, these results confirmed that IFA treatment facilitated effective host clearance of *K. pneumoniae* and thereby significantly controlled bacterial dissemination through targeting capsule.

Consistent with the severe airway inflammation observed via histopathology analysis, sublethal infection with wild-type K7 boosted the secretion of the inflammatory cytokines TNF-α, IL-6 and IL-1β, whereas the levels of these cytokines were significantly reduced by IFA treatment (Fig 8A–8C). Notably, IFA did not have any inhibitory effect on LPS-stimulated inflammation response in vitro (S4 Fig), indicating that IFA was unable to suppress TLR4-MyD88 signaling and that the protective effect of IFA in mice was independent of anti-inflammation activity. Somewhat surprisingly, the production of IL-10 and IFN-γ was much higher in mice infected with wild-type K7 and received IFA treatment (Fig 8D and 8E), indicating that IFA treatment facilitated the development of an inflammation-controlling and infection-limiting response to fight against *K. pneumoniae* in mice. Of note, the levels of these cytokines in mice infected with *ΔGT-1* K7 were all much lower than those in the wild-type K7-infected mice (Fig 8A-8E), which was consistent with effective bacterial clearance resulted from the deficient of

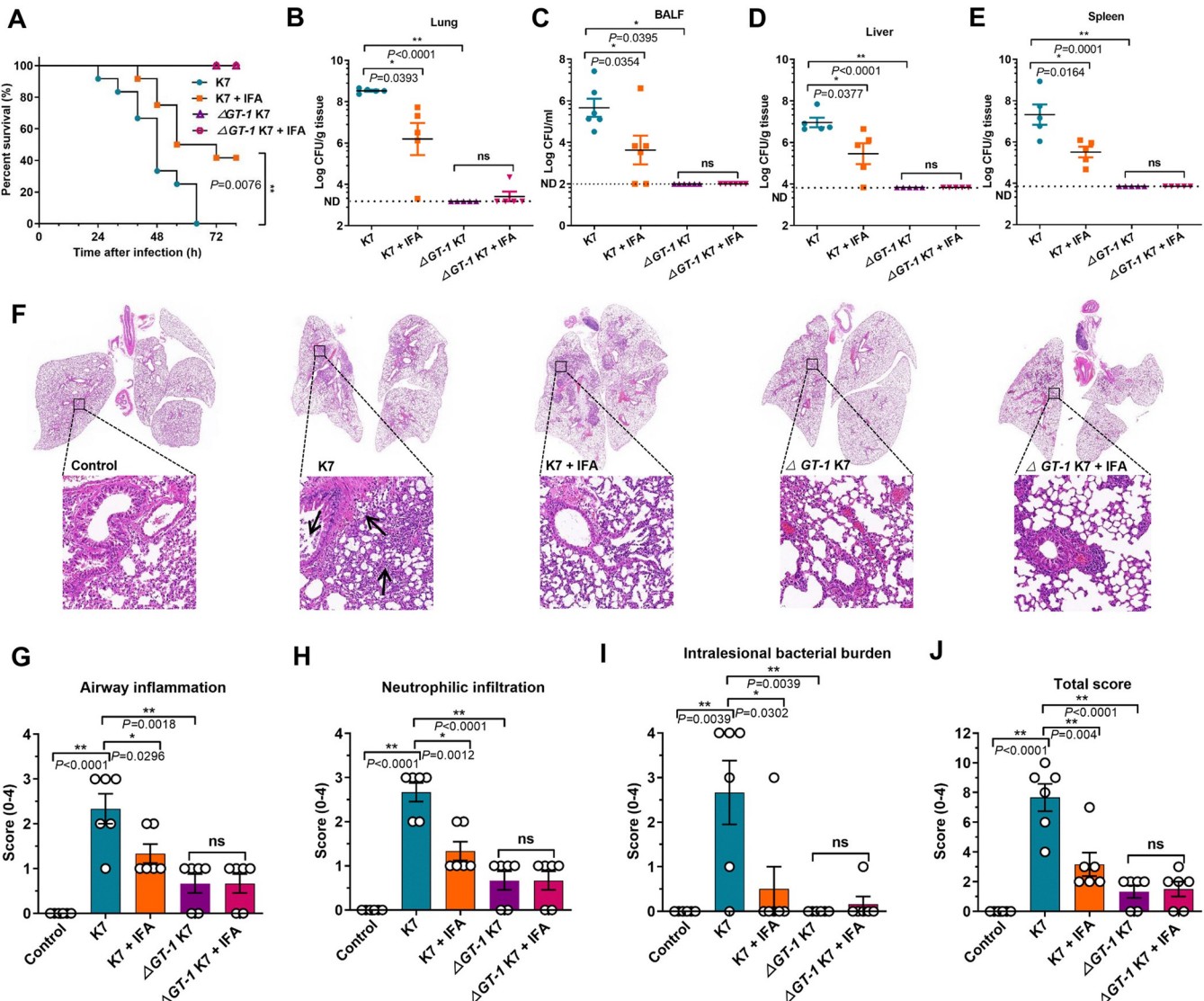

**Fig 7. IFA effectively limited bacterial dissemination and protected mice from lethal hypervirulent *K. pneumoniae* pneumonia.** (A) Survival of mice 72 hours after *K. pneumoniae* infection. Mice were subcutaneously injected with 50 mg/kg IFA in 50 μl of 10% DMSO containing vehicle (10% DMSO, 45% stroke-physiological saline solution, 40% PEG400 and 5% Tween-80) or an equal volume of vehicle immediately after challenge with $1 \times 10^{7}$ wild-type *K. pneumoniae* K7 or *ΔGT-1* K7 bacteria, and the number of deaths was recorded for survival analysis. Statistical analysis was performed using the log-rank (Mantel–Cox) test (n = 12 mice each group). (B) Bacterial burden in the lung tissues of sublethally infected mice. Mice challenged with $2.5 \times 10^{6}$ wild-type *K. pneumoniae* K7 or *ΔGT-1* K7 bacteria were treated as described above and sacrificed at 40 hours post infection, and lung tissues were removed and homogenized in sterilized PBS (10% w/v) to analyze the bacterial burden by microbiological plating (n = 5 mice each group). (C) Bacterial load in the bronchoalveolar lavage fluid (BALF) obtained from sublethally infected mice. Mice that were infected and treated as described above were sacrificed at 32 hours post infection for bronchoalveolar lavage (BAL) analysis, and the BALF was serially diluted and microbiologically plated (n = 6 mice in each group). The bacterial loads in the livers (D) and spleens (E) from the mice mentioned in (B) were also evaluated by microbiologically plating the tissue homogenates in PBS (n = 5 mice each group). ND, not detected. The back lines present the means ± SEMs. (F) H&E-stained lung tissues from mice infected and treated as indicated. The arrows indicate neutrophil infiltration. The degree of airway inflammation (G), neutrophilic infiltration (H), intralesional bacterial burden (I) and total histopathology score (J) were assessed according to standard pathology criteria by blinded scoring. The data are presented as the means ± SEMs. Unpaired two-tailed Student's t-test was performed to determine the statistical significance of two groups. *$P < 0.05$, **$P < 0.01$ and ns, no significance.

capsule. Consistently, the numbers of inflammatory monocytes (CD11b$^{+}$ Ly6C$^{high}$) and neutrophils (Cd11b$^{+}$Ly6G$^{+}$) in wild-type K7-infected mice were much greater than those in *ΔGT-1* K7-infected mice, but they were also markedly reduced by IFA treatment (Fig 8F and 8G).

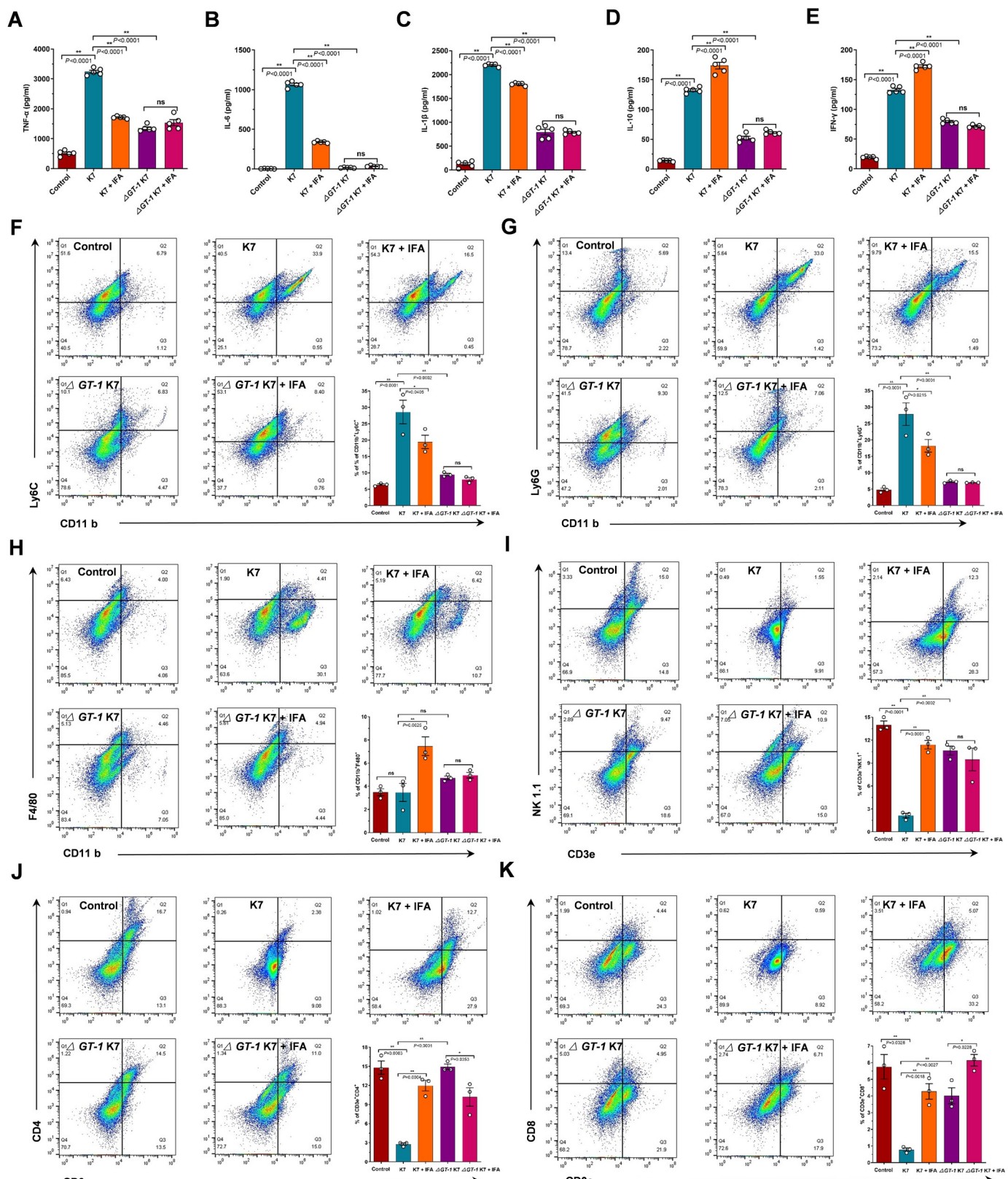

**Fig 8. IFA treatment facilitated host immune clearance of hypervirulent *K. pneumoniae* by targeting capsule.** Mice challenged with $2.5 \times 10^6$ wild-type *K. pneumoniae* K7 or *ΔGT-1* K7 bacteria were subcutaneously injected with 50 mg/kg IFA in 50 μl of 10% DMSO containing vehicle (10% DMSO, 45% stroke-

physiological saline solution, 40% PEG400 and 5% Tween-80) or an equal volume of vehicle, and were then sacrificed at 40 hours postinfection. The concentrations of TNF-$\alpha$(A), IL-6 (B), IL-1$\beta$(C), IL-10 (D) and IFN-$\gamma$ (E) in lung tissues were measured using ELISA kits following the manufacturer's instructions (n = 5 mice each group). To determine the number of immune cells in the BALF, mice infected and treated as indicated were sacrificed at 32 h postinfection for BALF collection. The populations of inflammatory monocytes (CD11b$^+$Ly6C$^+$) (F), neutrophils (CD11b$^+$Ly6G$^+$) (G), mature macrophages (CD11b$^+$F4/80$^+$) (H), NK cells (CD3e$^+$NK1.1$^+$) (I), CD4$^+$ T cells (CD3e$^+$CD4$^+$) (J) and CD8$^+$ T cells (CD3e$^+$CD8$^+$) (K) in the BALF from the indicated groups were quantified by flow cytometry (n = 3 mice each group). The y-axis in F-K means the percentage of gated 30000 cells from individual mice BALF for each marker set. The data are presented as the means ± SEMs. Unpaired two-tailed Student's t-test was performed to determine the statistical significance of two groups. $^*P < 0.05$, $^{**}P < 0.01$ and ns, no significance.

Conversely, compared with wild-type K7, the numbers of mature macrophages and NK cells were much greater in mice infected with *ΔGT-1* K7 (Fig 8H and 8I), which play essential roles in the defense against *K. pneumoniae* infection [29]. As speculated, IFA treatment effectively increased the number of these populations critical for host innate clearance. The populations of CD4$^+$ and CD8$^+$ T cells were also much lower in the mice infected with wild-type K7, but the numbers of these anti-infection adaptive immune cells were also significantly increased by IFA (Fig 8J and 8K), which was consistent with the increase in IFN-$\gamma$. The lower levels of IL-6, IL-1$\beta$ and TNF-$\alpha$ and populations of inflammatory monocytes and neutrophils and, in contrast, the greater populations of macrophages, NK cells, and CD4$^+$ and CD8$^+$ T cells in the lungs of mice treated with IFA explained how IFA provides systemic protection against *K. pneumoniae* infection in vivo.

Finally, we further distinguished the respective contribution of capsule and rmpD-regulated and hypercapsule-associated HMV to IFA-conferred protection against hvKP infection in vivo. Consistent with previous study, *rmpD* deletion resulted in a significant loss of virulence versus wild-type K7, as $1 \times 10^7$ CFU (a 100% lethal dose of wild-type K7) *ΔrmpD* K7 bacterial failed to cause death in mice (S2G Fig). Surprisingly, $5 \times 10^8$ CFU (50 fold of 100% lethal dose of wild-type K7) *ΔrmpD* K7 mutant still did not result in death within 72 hours (S2G Fig). Therefore, we performed bacterial burden assay by challenging $5 \times 10^7$ CFU bacterial (20 fold of the inoculation dose used before). As shown in S2H–S2J Fig, the bacterial burdens of *ΔrmpD* K7 were much lower compared with wild-type K7 but significantly greater than that of *ΔGT-1* K7, which confirmed the distinct contribution of HMV and capsule, as well as the indispensable role of capsule in hvKP pathogenicity. Unfortunately, although IFA treatment reduced the colonization of *Δrmp*D K7 in mice lungs by about 6 fold versus vehicle group (*P* = 0.0618) (S2H Fig), and by 4 fold in livers (*P* = 0.0716) (S2I Fig), neither of them achieved statistical significance. By contrast, bacterial loading in spleens was significantly reduced by IFA (*P* = 0.0211) (S2J Fig), indicating that the in vivo protection effect of IFA on hvKP infection is attributed to its inhibition on both capsule and hypercapsule-associated HMV, rather than acting on rmpD.

In conclusion, these results confirmed that IFA treatment promoted host immune clearance of *K. pneumoniae* by targeting capsule and hypercapsule-associated HMV, thereby systemically protecting mice against *K. pneumoniae* infection.

## Discussion

The plasmid delivery and mutual adaptation between plasmids and chromosomes of host bacteria endow *K. pneumoniae* with the ability to evolve toward increasing virulence and resistance levels. In particular, the global spread of hvKP further complicates clinical therapy for *K. pneumoniae* infections. Worse still, the successful convergence of carbapenem resistance and hypervirulence has created a new veritable superbug termed CR-hvKP, which poses a looming threat to public health owing to its high pathogenicity and increasing drug resistance. Although several new antimicrobial agents have been discovered in recent years, antibiotic

resistance is spreading faster than the introduction of new drugs into clinical use, and resistance can rapidly develop during treatment. Therefore, there is a constant need for novel alternative therapeutic drugs to curb bacterial infections, especially resistant infections. Several new therapeutic strategies aimed at addressing multidrug-resistant *Klebsiella* infections, such as host-directed therapy [30], antivirulence strategies [31], immunotherapy, bacteriophage cocktail therapy and phage-derived enzybiotics [32], have progressed recently, yet there are currently no available candidates for late-stage development.

Capsule is a critical virulence determinant for the full pathogenicity of most systemic pathogens capable of causing different infectious diseases, such as *K. pneumoniae* and other encapsuled pathogens, including *Acinetobacter baumannii*, *Neisseria meningitidis*, *Streptococcus pneumoniae* and *Staphylococcus aureus*. The important role of capsule in bacterial pathogenicity renders it a promising target for drug design. In the present study, we successfully identified IFA as a potent anti-capsule compound that effectively sensitized hvKP to host clearance and thus provided systemic protection against hvKP infection. In fact, capsule-based vaccines against *S. pneumoniae* have been successfully commercialized and widely used worldwide. Capsule polysaccharide-conjugated vaccines have also been shown to be effective against *K. pneumoniae* infection and have been successfully tested in preclinical and clinical studies, but no products are currently licensed or clinically available, because the diverse structure and great variability of *K. pneumoniae* capsule make it difficult for them to cross-react with different serotypes [33]. According to the awareness from epidemiological studies, only if the vaccine includes at least 25 capsular polysaccharides will it be able to cover approximately 70% of clinically relevant *K. pneumoniae* strains [4]. Moreover, capsule-targeted monoclonal antibodies also reduce the resistance of the clinical strain ST258 to complement-mediated serum killing and enhance phagocytosis and killing by human neutrophils [34], but serotype specificity is still the greatest challenge. Notably, capsule-targeted bacteriophages and phage-derived depolymerases also protect mice from *K. pneumoniae* infection [35,36], but their efficacy is strictly limited within host bacterial serotypes and even narrower owing to the diversity of capsular polysaccharide. The biosynthesis mechanism of *K. pneumoniae* capsule is highly conserved and therefore is advantageous as a target for the development of chemical inhibitors that stand out for their broad spectrum and low cost. Even though, we investigated the action mechanism of IFA through transcriptome analysis due to the highly complexity of the capsule synthesis system and regulatory factors, and accordingly, revealed that IFA reduced CPS biosynthesis through altering bacterial central metabolism, its specific action target needs to be further validated in the future. Particularly, applying of fluorescence- or chemiluminescence-based reporter systems may be an ideal high-throughput screening strategy for more anti-capsule drugs.

Hypermucoviscosity has been referred to as a characteristic phenotype of hvKP and is therefore often used as a distinguishing feature in clinical practice. Historically, hypermucoviscosity was deemed the result of abundant capsule production, yet emerging evidence revealed discordant changes in capsule and hypermucoviscosity phenotypically and genotypically [22,37], suggesting that the bacterial phenotype is a distinct important pathogenic factor involved in blocking adherence and internalization by host cells. But currently, research in this field still prove that the two essential virulence factors are always coordinated in most cases. Herein, we speculated that the inhibitory effect of IFA on hypermucoviscosity was attributed not only to decreased capsule production, because central metabolism and *K. pneumoniae* also directly impact hypermucoviscosity [23]; but also, to the significant alterations in carbon metabolism and the cellular redox status under IFA treatment. Notably, the broad-spectrum inhibitory effects of the small molecule on both capsule production and hypermucoviscosity in different serotype strains may also be attributed to changes in bacterial metabolism and energy

status. Moreover, an important observation was that the transcription levels of RcsB and RfaH, the two major positive regulators of cps locus, both were significantly suppressed by IFA, indicating that IFA might act at upstream of cps gene expression. These findings elucidated the prospects of IFA for inhibiting two critical distinct pathogenic phenotypes independent of serotype and the diversity of capsular polysaccharide.

Although, how capsule contributes to bacterial pathogenicity during infection remains unknown, its major function is to mediate immune evasion and impede innate clearance of hvKP or cKP, which is pervasive throughout the literature related to *K. pneumoniae*. Notably, capsule-conferred resistance to the complement system is strongly associated with systemic survival and rapid spread of hvKP and is therefore a vital mechanism that facilitates lethal invasive infection in hvKP [38]. In this study, we found that IFA significantly decreased the survival ratio of hvKP in human serum, which could partially explain the effective control of bacterial dissemination at the early stage of hvKP infection. A previous study emphasized that HMV is more critical for hvKP to block adherence, and both capsule and HMV are required for the prevention of macrophage phagocytosis [22]. Interestingly, our results elucidated that IFA also improved the adherence of *ΔrmpD*-K7, a mutant maintained hypercapsule but deficient in HMV, indicating that the contribution of rmpD and capsule in bocking adherence might be discrepant between different serotypes or strains. Moreover, regardless of that acapsular strains have similar intracellular survival within macrophages compared to WT strains [12], IFA-caused significant increase in phagocytosis might be critical for its in vivo effectiveness. In addition, although IFA reduced the colonization of *ΔrmpD*-K7 in mice to some extent, it failed to achieve statistical significance similarly to wild-type strain, demonstrating that the therapeutic effect of IFA on hvKP infection depends on both capsule and hypercapsule-associated HMV. Together, our study provided comprehensive evidences of how IFA curbs hvKP infection from both the perspective of the capsule itself and the HMV phenotype mediated by the capsule and rmpD.

IFN-γ plays a critical role in the defense against bacterial infections as an indispensable bridge between innate and adaptive immunity [39], our result confirmed that the level of IFN-γ and the number of CD4$^+$ and CD8$^+$ T cells were significantly improved by IFA; consequently, the bacterial burden in mice lungs and bacterial dissemination from lung to other organs were effectively controlled. In addition, IFA treatment significantly increased the level of IL-10, while decreased the levels of inflammatory cytokines and cells in the lung tissue, which was in accordance with the alleviated inflammatory tissue injury. These results further confirmed the mechanisms underlying how IFA facilitated host clearance against *K. pneumoniae* through targeting capsule and clearly demonstrated that IFA treatment facilitated the development of an inflammation-controlling and infection-clearance immune response to combat *K. pneumoniae*.

Overall, in this study, IFA was identified as a potent inhibitor of the *K. pneumoniae* capsule that accelerates effective host clearance through blunting capsule-mediated immune evasion strategies. Notably, we propose that the combination of IFA with several newly discovered host-directed drugs, which are designed to combat infection through disrupting the host cellular events required for bacterial survival or multiplication or augmenting host immune responses, such as the bioactive lipid phosphatidylinositol 5-phoshate (PI5P) [40], and autophagy-inducing drug bromhexine hydrochloride [41], is a highly desirable strategy to improve the poor clinical outcomes of hvKP infection and, more importantly, to effectively circumvent antibiotic resistance. The findings of this study figure out a successful avenue to curb hvKP infections, as well as CRKP and CR-hvKP infections, and shed light on the development of capsule-targeted drugs.

## Materials and methods

### Ethics statement

All the animal experiments were approved and strictly conducted in compliance with the relevant guidelines established by the Jilin University Institution Animal Care Committee (approval no. SY202412069).

### Reagents and antibodies

IFA (>98%) purchased from Shanghai Yuanye Biotechnology was dissolved in dimethyl sulfoxide (DMSO; Sigma–Aldrich) to make a 40 mg/mL stock solution for in vitro experiments. For in vivo delivery, IFA was dissolved in a vehicle solvent consisting of 10% DMSO, 45% stroke-physiological saline solution, 40% PEG400 and 5% Tween-80 to make a 40 mg/mL stock for injection unless otherwise indicated. Apramycin, spectinomycin, L-arabinose and sucrose used for colony screen were from Shanghai Aladdin Biotechnology. Zwittergent 3–14, citric acid and 3-hydroxydiphenol used for uronic acid measurement were purchased from Sigma–Aldrich. ATP Assay Kit and Enhanced NAD+/NADH Assay Kit with WST-8 were purchased from Beyotime. Lipopolysaccharide from *E. coli* O26:B6 was purchased from Sigma–Aldrich. Mouse IL-1β, IL-6, TNF-α, IL-10 and IFN-γ kits from Biolegend were used to detect cytokines. A mouse monoclonal allophycocyanin (APC) anti-C3b/iC3b antibody (BioLegend) was used for the detection of C3b/iC3b deposition. The mouse anti-C5b-9 antibody and Alexa Fluor 488-conjugated goat anti-mouse IgG H+L used for the detection of the C5b-9 load were both obtained from Abcam. The following antibodies from BD Biosciences were used for the staining of neutrophils, monocytes, NK cells, mature macrophages, CD4$^+$ T cells and CD8$^+$ T cells in the bronchoalveolar lavage fluid (BALF): rat anti-mouse CD16/CD32, anti-CD11b-APC, anti-Ly6G-PE-Cy7, anti-Ly6C-FITC, anti-CD3e-Percy-Cy5, anti-NK1.1-APC, anti-F4/80-PE, anti-CD4-APC-Cy7 and anti-CD8-BV421.

### Bacterial strains, culture conditions and plasmids

The clinical isolation of carbapenem-sensitive K2-serotype hypervirulent *K. pneumoniae* strain K7 and other isolates, including WKP5 (K2), WKP22 (K2), WKP11 (K1), WKP13 (K1), WKP25 (K1), WKP35 (K57), KPP29 (K57) and WKP50 (K61) gifted from Gu's lab, and hvKP ATCC43816 (K2) gifted from Zhang's lab were all cultured in Luria–Bertani (LB) broth with shaking at 37˚C. Notably, the capsule deficient mutant *ΔGT-1* K7 from Gu was constructed as previously described and cultured in Luria–Bertani (LB) broth containing 30 μg/ml kanamycin [42]. For comparative assessment of colony mucoidity, isolates were plated on LB agar or blood agar plates (LB agar with 5% defibrinated sheep blood). PCasKP-apr plasmid and pSGKP-spec plasmid gifted from Ji's lab were used to generate a *rmpD*-deletion K7 mutant.

### Construction of a *rmpD*-deletion mutant

*RmpD*-deletion K7 was generated using a CRISPR-based genome-editing approach [43]. In brief, designed single artificial chimeric guide RNA (sgRNA) was ligated into the linearized pSGKP-spec plasmid, and the linear double-stranded DNA (dsDNA) homologous arms of *rmpD* gene was amplified by fusion PCR as a repair template. Electroporate no less than 200 ng pCasKP-apr plasmid into competent K7 cells to prepare pCasKP-harboring competent K7 cells. Subsequently, 200 ng spacer-induced pSGKP-sgRNA plasmid and 500 ng dsDNA repair template were co-transformed into pCasKP-harboring competent K7 cells by electroporation. Positive transformants were screened using 30 μg/mL apramycin and 300 μg/mL spectinomycin at 30˚C and verified by DNA sequencing. Last, plasmids were cured by culturing the single

mutant colony at 37˚C in the presence of 5% sucrose. Primers used in the in-frame deletion and verification are listed in S1 Table.

## Cell lines and mice

Human A549 alveolar epithelial cells and mouse J774 macrophages were cultured in DMEM (Dulbecco's modified Eagle's medium, Gibco) supplemented with 10% fetal bovine serum (FBS) and 1% penicillin/streptomycin in a 5% $CO_2$ incubator. Mouse primary peritoneal macrophages were collected as previously described and cultured in RPMI 1640 (Roswell Park Memorial Institute 1640 Medium) supplemented with 10% FBS [44].

Eight-week-old female C57BL/6 mice were purchased from Changsheng Biotechnology (Shenyang, China) and humanely housed in individually ventilated cages (IVCs) under standard laboratory conditions (25 ± 1˚C, 12-h light/dark cycle with free access to food and water).

## Extraction and quantification of capsule polysaccharide (CPS)

The extraction and quantification of CPS were performed according to a modified procedure [45]. In brief, overnight *K. pneumoniae* cultures were transferred to freshly prepared LB broth containing DMSO or the indicated concentrations of IFA at a ratio of 1:50 and further cultivated for 4 hours with shaking at 80 rpm. To determine the total CPS production, 1 ml subcultures were directly mixed with 200 μl of 1% Zwittergent 3–14 detergent in 100 mM citric acid (pH 2.0) and incubated at 50˚C for 20 min to extract CPS. To measure cell-attached and unattached CPS, 1 ml aliquots were centrifuged at 13,000 × g for 5 min. The supernatants and pellets resuspended in 1 ml of fresh LB broth were also mixed with 200 μl of 1% Zwittergent 3–14 detergent and incubated under the same conditions. Then, all the aliquots were centrifuged at 13,000 × g for 5 min, and the CPS in the supernatants was precipitated with ethanol (80% final concentration) at 4˚C for 20 min and air-dried after discarding the supernatants. Finally, the CPS samples were dissolved in tetraborate/sulfuric acid, developed with 3-hydroxydiphenol, and quantified according to the standard curve of glucuronic acid based on the absorbance of the mixture at 520 nm. In addition, CPS samples extracted from equal amounts of *K. pneumoniae* strains were separated by 12% SDS-PAGE and followed by alcian blue staining, visualization and quantification by Image J.

## Scanning electron microscopy (SEM) and transmission electron microscopy (TEM)

The subcultures of *K. pneumoniae* described above were washed 3 times with sterilized PBS and fixed in 4% glutaraldehyde overnight at 4˚C. The immobilized samples were dehydrated by gradient ethanol (20, 50, 70, 80, 90, 95 and 100) and then dried under flowing nitrogen on cover glasses. Finally, bacterial morphology was observed by ZEISS scanning electron microscopy (SEM) at an acceleration voltage of 2 kV, and images were captured at a magnification of 15,000 ×.

For TEM, the fixed samples were embedded in Lowicryl HM20 Monostep resin and cut into ultrathin sections on an ultramicrotome, followed by staining with uranyl acetate and lead citrate. Images were obtained on an FEI Tecnai Spirit 120 kV transmission electron microscope with a Tietz F4.15 CCD camera.

## Susceptibility assay

The minimum inhibitory concentration (MIC) of IFA on *K. pneumoniae* was determined by the broth microdilution method [46]. The growth curves of *K. pneumoniae* with or without IFA were determined by spectrophotometry as previously described [47].

## Comparative assessment of colony mucoidity

*K. pneumoniae* strains were serially diluted and plated for fully separated, single colonies on blood agar plates or regular agar plates containing DMSO or 32 μg/ml IFA. After 18 hours of cultivation at 37°C, colony morphology was captured by photography, and the "string test" was performed as previously described [48].

## Centrifugation resistance analysis

A semiquantitative sedimentation assay was performed as follows: triplicate aliquots (1 ml) of the subcultures cocultured with DMSO or the indicated concentrations of IFA were centrifuged at $1,000 \times g$ in a fixed angle rotor for 5 min after the original $OD_{600}$ was measured by spectrophotometry, following which the supernatants were removed and transferred into new tubes to measure the $OD_{600}$. The results were defined as the ratio of the supernatant $OD_{600}$ to the original $OD_{600}$.

## RNA-seq analysis

RNA sequencing was carried out with 3 replicates to quantify patterns of gene expression in *K. pneumoniae* treated with DMSO control or 32 μg/ml. Total RNA was extracted using an AMPure XP system (Beckman Coulter, Beverly, USA) according to the manufacturer's instructions. RNA-seq was performed on an Illumina Novaseq platform (Beijing, China). Differential expression analysis of two groups (two biological replicates per group) was performed using the DESeq R package (1.18.0). Quantitative analysis of gene expression, differential gene expression analysis, and Gene Ontology (GO) analysis were subsequently performed to explore and identify enriched functions and pathways of differentially expressed genes. Notably, GO enrichment analysis of differentially expressed genes was implemented by the GOseq R package. Type-1 errors were reduced by performing false discovery rate (FDR) correction via Benjamini and Hochberg (BH) methods. GO terms and pathways with adjusted p values $\leq 0.05$ were significantly enriched in DEGs after multiple testing correction.

## Quantitation of intracellular ATP and the ratio of NAD+/NADH

Overnight *K. pneumoniae* cultures were transferred to freshly prepared LB broth containing DMSO or the indicated concentrations of IFA at a ratio of 1:50 and further cultivated for 4 hours with shaking at 80 rpm. Subsequently, equal amounts of bacterial were collected by centrifugation at $12,000 \times g$ for 5 min. The intracellular ATP levels and the ratio of NAD+/NADH were then analyzed using Enhanced ATP Assay Kit and Enhanced NAD+/NADH Assay Kit with WST-8, respectively.

## Serum survival assays

Overnight cultures of *K. pneumoniae* were subcultured in fresh LB broth supplemented with DMSO or the indicated concentrations of IFA for 4 hours. One milliliter of each culture aliquot was centrifuged, washed once with sterilized PBS, resuspended in 1 ml of PBS and then diluted at a ratio of 1:100. Two hundred microliters of the dilutions were then incubated with 20% (final concentration) prewarmed normal human serum (NHS) in a 37°C incubator with rotation. At each indicated time point, 20 μl aliquots from each assay tube were immediately serially diluted and plated on LB agar for colony enumeration.

## Detection of surface-deposited complement components

The subcultures described above were collected by centrifugation and sensitized in 1 ml of $GVB^{2+}$ buffer (0.15 mM $CaCl_2$, 141 mM NaCl, 0.5 mM $MgCl_2$, 0.1% gelatin and 5 mM veronal) for 5 min at room temperature. Two hundred microliters of the 1:100 diluted suspensions were incubated with 20% NHS in a 37°C incubator with rotation for the indicated times. Then, the mixtures were centrifuged and washed with PBS. C3b/C3bi was directly labeled with a mouse anti-C3b/C3bi-APC antibody (4 μl per $10^6$ bacterial cells), and C5b-9 membrane attack complex formation was indirectly detected with a mouse anti-C5b-9 antibody as the primary antibody (2 μg/ml) and Alexa Fluor 488 goat anti-mouse IgG H+L as the secondary antibody (1:1000). After 20 minutes of incubation at room temperature, the samples were washed 3 times with PBS and resuspended in 220 μl of PBS. Twenty microliters of each aliquot were transferred to cover glasses for image capture on a laser scanning confocal microscope (Olympus, Japan), and the remaining samples were immediately acquired using a BD FACSCelesta instrument. Approximately 10,000 cell events were collected and analyzed using FlowJo 10 software. Notably, a serum-free control was set to exclude non-specific binding of the secondary antibody. The immunofluorescence intensity was quantified using Image J software. Graph design and statistical analysis were performed using GraphPad 8.4.2 software.

## Adhesion and macrophage phagocytosis assay

Adhesion and macrophage phagocytosis assay were performed according to a previously study with some modification [48]. A549 cells were seeded in 24-well plates at a density of $2 \times 10^5$ cells per well and cultured overnight in a 5% $CO_2$ incubator. Overnight cultures of *K. pneumoniae* were subcultured in fresh LB broth supplemented with DMSO or the indicated concentrations of IFA for 4 hours and then added to the cell wells at an MOI of 50, followed by centrifugation at $1,000 \times g$ for 10 min to synchronize infection. After a 2-hour incubation, the monolayers were washed 3 times with PBS and lysed with 0.1% Triton-X 100. The lysates were serially diluted and plated on LB agar to determine the adherence efficiency by enumerating the colony-forming units. The data are presented as a percentage of the initial inoculum CFU.

J774 cells and mouse primary peritoneal macrophages (MPMs) were seeded in 24-well plates at a density of $5 \times 10^5$ cells per well and cultured overnight in a 5% $CO_2$ incubator. Cells were then infected with *K. pneumoniae* at an MOI of 5 with different treatments and centrifuged at $1,000 \times g$ for 10 min to synchronize infection. After being incubated for 2 hours, the cells were rinsed 3 times with PBS and further incubated for indicated times in fresh medium containing 100 μg/ml gentamicin to kill extracellular bacteria. The cells were then rinsed, lysed, serially diluted and plated on LB agar to enumerate the colony-forming units. The phagocytosis percentage of each inoculum was defined as the ratio of phagocytized bacteria to the total bacterial count determined before adding gentamicin.

## RNA extraction and RT–qPCR

The subcultures described above were collected at log-phase for the detection of transcriptional levels of cps genes and rmp genes. Total RNA was isolated using the TRIzol method. The RT–qPCR primer pairs used in the study are listed in S1 Table. RT–qPCR was carried out on an Applied Bioscience 7500 thermocycler using FastStart Universal SYBR Green Master Mix (Gene Star) according to the manufacturer's protocol [49].

### *Galleria mellonella* infection

Larvae of *G. mellonella* were purchased from Yude Biotechnology Ltd (Tianjin, China). Overnight cultures of *K. pneumoniae* were subcultured at 1:100 in fresh LB broth to an $OD_{600}$ of approximately 0.6. Bacteria were collected, washed 3 times with sterilized PBS and resuspended in PBS. Ten microliters of the suspension containing $2 \times 10^4$ bacterial cells or PBS were injected into the foremost right-side proleg of the larvae using a microinjection pump, and 10 μl of vehicle or IFA was delivered via the foremost left-side proleg at a dose of 50 mg/kg. Larvae in the control group were injected with 10 μl of PBS and vehicle. After injection, the larvae were moved to a 37°C incubator, and the number of deaths was scored periodically. To determine the bacterial load in the hemocoel of the larvae, the larvae were placed on ice for 20 min at the fifth hour after infection. The bottom 2 mm of the larvae was aseptically cut off, and the hemocoel was collected into 1.5 ml tubes, followed by serial dilution and microbiological plating. Moreover, melanization and lactate dehydrogenase (LDH) assays were performed as described previously [28]. Of note, in LDH assay, IFA was dissolved in 1% DMSO containing vehicle (1% DMSO, 54% stroke-physiological saline solution, 40% PEG400 and 5% Tween-80) rather than 10% DMSO. All experiments were carried out at least in triplicate.

### Murine pneumonia model

For the analysis of mouse mortality, female C57BL/6 mice (6–8 weeks old) were challenged with $1 \times 10^7$ CFU of K7 or *ΔGT-1* K7 bacteria via the intranasal route, followed by subcutaneous injection of a single dose of 50 mg/kg IFA dissolved in 10% DMSO containing vehicle (10% DMSO, 45% stroke-physiological saline solution, 40% PEG400 and 5% Tween-80) or an equal volume of vehicle (50 μl) immediately. IFA and vehicle were administered 3 times daily, and the number of dead mice was counted until 72 hours after inoculation. To assess the histopathology of the lung tissues, mice were infected with $2.5 \times 10^6$ CFU of bacteria, treated with IFA or vehicle in the same way, anesthetized with isoflurane and euthanized through rapid cervical dislocation at 40 hours post infection. The mouse lungs were fixed in 10% formalin, stained with hematoxylin and eosin (H&E) and visualized for histopathological examination. Standard pathology criteria were applied to score the histopathology features of the lung lesions [29]. Additionally, the lungs, livers and spleens from infected mice that were treated as described above were homogenized in sterilized PBS to prepare a 10% homogenate (w/v) for the analysis of the bacterial load via microbiological plating. The cytokine levels in the supernatants of lung tissue homogenates were assessed using ELISA. Bronchoalveolar lavage (BAL) was carried out at 32 hours post infection according to a previous study [50], and bacterial loads in BALF were also determined. Total cells in the BALF were collected by centrifugation at $600 \times g$ for 5 min, and erythrocytes were then removed with 1 ml of RBC lysis buffer. To further verify the respective contribution of capsule and HMV in IFA-conferred protection, mice were challenged with $5 \times 10^7$ CFU K7, *ΔGT-1* K7 and *ΔrmpD* K7 for bacterial burden analysis.

### Staining and analysis of innate and adaptive immune cells by flow cytometry

The cells collected from the BALF described above were resuspended in 1 ml of PBS supplemented with 0.5% BSA, passed through a 70-μm strainer and incubated with a mouse anti-CD16/32 antibody at a dilution of 1:100 to block Fc receptors. After 10 min of incubation at room temperature and washing once with PBS containing 0.5% BSA, half of the cells were transferred to new tubes and further incubated with anti-CD3e-Percy-Cy5, anti-NK1.1-APC, anti-CD4-APC-Cy7 and anti-CD8-BV421 antibodies at 4°C for 30 min, whereas the

remaining cells were stained with anti-CD11b+-APC, F4/80-PE, anti-Ly6G-PE-Cy7 and anti-Ly6C-FITC. All the cells were then washed once and resuspended in 300 μl of PBS containing 0.5% BSA. A total of 30,000 cell events were detected on a BD FACSCelesta instrument and analyzed using FlowJo software. The populations of neutrophils (CD11b$^+$Ly6G$^+$), inflammatory monocytes (CD11b$^+$Ly6C$^+$), macrophages (CD11b$^+$F4/80$^+$), NK cells (CD3e$^+$NK1.1$^+$), CD4$^+$T cells (CD3e$^+$CD4$^+$) and CD8$^+$T cells (CD3e$^+$CD8$^+$) were gated. Unless otherwise indicated, all percentages are of single viable frequency, and the results from 3 mice were analyzed using GraphPad Prism 8.4.2.

## Statistical analysis

Statistical analysis was performed using GraphPad Prism 8.4.2 (USA). The numeric data from no less than three independent experiments are presented as the mean ± standard error of the mean (SEM). The statistical significance of differences between two independent groups was determined by unpaired two-tailed Student's t test. Comparisons of 3 or more data points were analyzed using one-way analysis of variance (ANOVA) and Tukey's posttest, as indicated. The survival rate was analyzed using the log-rank (Mantel–Cox) test. $^*P < 0.05$, $^{**}P < 0.01$ and ns, no significance.

## Supporting information

**S1 Fig. Uronic acid analysis-based screen identified IFA as an effective capsule inhibitor.** (A) Diagram for compound screening of capsule inhibitors. (B) Percentage inhibition of uronic acid levels by each compound tested at 32 μg/ml.
(TIF)

**S2 Fig. Validation of distinct contribution of capsule and hypercapsule-associated HMV in IFA-conferred protection against hvKP infection using a *ΔrmpD*-K7 mutant.** (A) The deletion of the *rmpD* gene in K7 strain. The PCR band from wild type strain served as a positive control. (B) The semiquantitative sedimentation assay of wild type strain and *ΔrmpD*-K7 mutant. (C) The colony morphology observation and "string test". (D) Determination of capsule production by uronic acid assay. Total uronic acid samples were extracted from *ΔrmpD*-K7 mutant co-cultured with indicated concentrations of IFA. (E) Adhesion assay of *K. pneumoniae* K7 and *ΔrmpD* mutant to A549 cells in the presence of DMSO or indicated concentrations of IFA. Cells were infected with *K. pneumoniae* strains at an MOI of 50 for 2 hours and then washed, lysed and plated on LB agar plates to quantify the colony-forming units. Data are presented as a percentage of the initial inoculum CFU. (F) Phagocytosis of *K. pneumoniae* K7 and *ΔrmpD* mutant by J774 macrophages in the presence of vehicle or indicated concentrations of IFA. Cells were infected at an MOI of 5 for 2 hours and then washed, followed by further 1-hour incubation in the medium containing gentamicin (100 μg/ml) to kill extracellular bacteria. The cells were then rinsed, lysed and plated on LB agar plates after serial dilution. (G) Survival of mice challenged by $1 \times 10^7$ (n = 6 mice each group) or $5 \times 10^8$ *ΔrmpD* K7 (n = 8 mice each group) with indicated treatments. Mice were subcutaneously injected with 50 mg/kg IFA in 50 μl of 10% DMSO containing vehicle (10% DMSO, 45% stroke-physiological saline solution, 40% PEG400 and 5% Tween-80) or an equal volume of vehicle immediately after infection, and the number of deaths was recorded for survival analysis. (H) Bacterial burden in the lung tissues of mice infected with *K. pneumoniae* strains. Mice challenged with $5 \times 10^7$ wild-type K7, *ΔGT-1* K7 and *ΔrmpD* K7 bacteria were treated as indicated and sacrificed at 40 hours post infection, and lung tissues were removed and homogenized in PBS to analyze the bacterial burden by microbiological plating (n = 5 mice each group). The bacterial loads in the

livers (I) and spleens (J) were also evaluated by microbiologically plating the tissue homogenates in sterilized PBS (10% w/v) (n = 5 mice each group). ND, not detected. The back lines present the means ± SEMs. Comparisons in (D) were analyzed by one-way ANOVA and Tukey's posttest, other comparisons between two groups were calculated by unpaired two-tailed Student's t-test. $^*P < 0.05$, $^{**}P < 0.01$ and ns, no significance.
(TIF)

**S3 Fig. IFA significantly attenuated capsule-mediated resistance to the binding of C3b and C5b-9 to bacterial surface.** (A) The immunofluorescence intensity of C3b/C3bi quantified by Image J. (B) The immunofluorescence intensity of C5b-9 quantified by Image J. The mean ± SEM is shown. Data were analyzed using unpaired two-tailed Student's t-test. $^{**}P < 0.01$ and ns, no significance.
(TIF)

**S4 Fig. IFA did not affect LPS-induced inflammation response.** To determine the effect of IFA on the TLR4 ligand LPS-stimulated inflammation response, mouse primary peritoneal macrophages (MPMs) were stimulated with LPS (1 mg/mL) for 18 hours in the presence of DMSO or the indicated concentrations of IFA, and the levels of IL-6 (A) and TNF-$\alpha$ (B) in the culture supernatants were detected using ELISA. The data are presented as the means ± SEMs. Data were analyzed by one-way ANOVA and Tukey's posttest. $^{**}P < 0.01$ and ns, no significance compared with DMSO control.
(TIF)

**S1 Table. Sequence of primer pairs used in the study.**
(DOCX)

## Acknowledgments

We would like to gratefully acknowledge Professors Jingmin Gu, Jingren Zhang and Quanjiang Ji for providing clinical isolates, ATCC43816 strain and the pCasKP-pSGKP two-plasmid bacterial genome editing system, respectively.

## Author Contributions

**Conceptualization:** Tingting Wang, Huaizhi Yang, Jianfeng Wang, Lei Song, Xuming Deng.

**Data curation:** Tingting Wang, Qiushuang Sheng.

**Formal analysis:** Lei Song, Xuming Deng.

**Funding acquisition:** Lei Song, Xuming Deng.

**Investigation:** Huaizhi Yang, Feng Chen.

**Methodology:** Tingting Wang, Huaizhi Yang, Qiushuang Sheng, Ying Ding, Jian Zhang, Feng Chen, Lei Song.

**Project administration:** Tingting Wang.

**Resources:** Lei Song.

**Software:** Qiushuang Sheng, Ying Ding.

**Supervision:** Jianfeng Wang, Lei Song, Xuming Deng.

**Validation:** Huaizhi Yang, Qiushuang Sheng.

**Visualization:** Tingting Wang, Huaizhi Yang, Ying Ding.

**Writing – original draft:** Tingting Wang.

**Writing – review & editing:** Tingting Wang, Jianfeng Wang, Lei Song, Xuming Deng.

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
