## [Decision Letter · Decision Letter 0]

3 Aug 2024

Dear Dr. Deng,

Thank you very much for submitting your manuscript "Isoferulic acid facilitates effective clearance of hypervirulent Klebsiella pneumoniae through targeting capsule" for consideration at PLOS Pathogens. Your manuscript was reviewed by members of the editorial board and by three independent reviewers. In light of the reviews (below), we would like to invite the resubmission of a substantially revised version that takes into account the reviewers' comments. Please, pay particular attention to the concerns raised with regards to streamlining of the manuscript, construction and testing of a rmpD mutant (which will require considerable additional experimentation), proper citations, and concerns regards IFA as vehicle control. Note that we suggest that your focus your study on the drug screen that identified IFA-targeted capsule inhibition; the manuscript is weighed down by the experiments on the mechanism of innate immune responses and TLR4, epithelial cells and macrophages, which does not really contribute any novel insight into capsule interactions with the innate immune system.

We cannot make any decision about publication until we have seen the revised manuscript and your response to the reviewers' comments. Your revised manuscript is also likely to be sent to reviewers for further evaluation.

Sincerely,

M Ammar Zafar

Guest Editor

PLOS Pathogens

D. Scott Samuels

Section Editor

PLOS Pathogens

Michael Malim

Editor-in-Chief

PLOS Pathogens

orcid.org/0000-0002-7699-2064

Reviewer's Responses to Questions

**Part I - Summary**

Reviewer #1: In this study by Wang and colleagues, the authors identify the compound isoferulic acid (IFA) as an inhibitor of Klebsiella pneumoniae capsule production. Using a combination of in vitro and in vivo techniques, the authors define IFA inhibition of K. pneumoniae capsule and seek to determine how this inhibition influences bacterial pathogenicity and host responses. One of the great strengths of this study is the utilization of several techniques to confirm early key findings. For example, uronic acid quantification, SEM, centrifugation assays, and the string test are all used to characterize and differentiate capsule and hypermucoviscosity. While the initial characterization of IFA in capsule inhibition is quite interesting, I have major concerns about much of the characterization of IFA in host immune responses. In many instances statistical comparisons are made between irrelevant groups, and some data implies that the IFA vehicle itself is modulating host responses. Additionally, the influence of K. pneumoniae capsule on host immune responses has been previously extensively characterized and I am not convinced that this study is providing novel insight into capsule biology.

Reviewer #2: This manuscript by Wang, Yang et al. reports the findings of a study exploring the effects of isoferulic acid (IFA) on Klebsiella pneumoniae (Kp) capsule expression and resulting phenotypes. The authors should be commended for their extensive experimental approach. This greatly increases the confidence in the authors' conclusion that IFA represses capsule to a degree that impacts the pathogenesis of Kp. Given that the role of capsule in Kp virulence is very well studied, much of data are supporting, rather than discovery; however, this does increase the confidence in the conclusions regarding the impact of IFA on Kp. Unfortunately, the in vivo regiment for IFA delivery is very intense (3 injections/day), potentially hindering the potential of IFA as a therapeutic. Thus, the authors conclusions regarding the therapeutic use of IFA may be overstated. There are many areas where the manuscript can be clarified, and it would benefit from another round of careful editing.

Reviewer #3: Wang, et al. (PPATHOGENS-D-24-01091) report a series of experiments showing that treatment of hypervirulent Klebsiella pneumoniae cultures with a newly-identified compound called IFA reduces capsule and hypermucoviscosity (HMV). These phenotypic changes render the bacteria more sensitive to complement binding and killing, less virulent in both Galleria and mouse models of infection, and altered immunological responses by the host. Given the growing threat of antibiotic resistant Klebsiella, new therapeutics are much needed and the data presented here have value to the community. The strengths of this manuscript lie in the vast number of experiments showing that IFA treatment improves host survival, most likely by reducing capsule production/HMV. The organization of the data is logical and the experiments seem to be well-executed with appropriate controls. However, the vast number of experiments is also a weakness! I found myself exhausted by the volume of data and by the challenges of examining many small data panels that were insufficiently labeled. Other weaknesses stem from overstated conclusions and a lack of clarity with some experiments. Most comments provided below are in an effort to make this manuscript more readable.

Note: the data presented in Figures 5, 9 and 10 are outside my scientific expertise.

**Part II – Major Issues: Key Experiments Required for Acceptance**

Reviewer #1: Major Concerns:

1. The field of Klebsiella pathogenesis has extensively characterized the impact of capsule in pathogenicity and the authors state that capsule is one of K. pneumoniae’s most relevant virulence factors. For example, in previous studies authors have determined capsule to be important for in vivo virulence (PMID16262790), serum susceptibility (PMID26060277), and TLR4-mediated inflammation (PMID25971969). These references, and many others, have extensively established that K. pneumoniae capsule helps to protect the bacteria from innate immune threats. I am highly enthusiastic about the author’s findings in Figures 1-3, but am not convinced that the authors have uncovered novel biology in latter figures as these results largely confirm previous findings. The study would be stronger if the paper’s focus was solely the suppression of capsule by IFA in Figures 1-3, and not focused on the role of capsule in pathogenicity.

2. The statistical comparisons in the latter part of the manuscript are not always appropriate for making the conclusion that IFA-mediated inhibition of K7 capsule is modulating the immune response. In many panels, the authors neglect to show statistical differences between the WT and acapsular K7 strains (Figure 4F, 4H). This is important for first demonstrating that lack of capsule is important for the methodology, then determining that a WT+IFA comparison has a similar result. In Figure 5A&5B, a comparison between K7 and acapsular K7 would likely show no difference in expression of the gene is interest, so it is curious why the IFA administration significantly increases expression in many cases. The authors report these comparisons in 5C&5D, the approach should be consistent in other panels throughout the manuscript.

3. Perhaps my biggest concern is the possibility that the IFA vehicle is substantially modulating immune responses. For example, in Figure 5 there are many panels in which the WT and acapsular K7 strains elicit similar expression of the genes of interest. However, there is a substantial increase in expression of certain genes when IFA is added (ex. Figure 5A, B, F). Again, in Figure 7D the authors claim that acapsular K. pneumoniae elicits less LDH release than the WT strains (Lines 333-334) but this statistical comparison is not made in the corresponding graph. By eye, it appears that the WT and acapsular K7 strains elicit quite similar LDH release. Since the statistical comparison is then significant between the K7 and the K&+IFA strains, this implies that the IFA vehicle could be influencing LDH release. To underscore this point, the control in the LDH experiment is a PBS control, not the vehicle in which IFA is administered. Another example can be found in Figure 6B, where there is a substantial change in defensin1 expression when the acapsular K7 is treated with IFA. Another instance of potential vehicle effects is in animal modeling. In Figure 7C, the Galleria treated with K7+IFA have substantially higher OD490 than the K7 or the acapsular strains.

4. The authors do an excellent job of defining IFA as a compound that suppresses capsule, but the study itself should be streamlined to emphasize the most important points. It is unclear whether the authors view their major finding as the identification of IFA-capsule interactions, characterizing capsule in innate immune responses, or capsule-specific engagement of TLR4 pathways. The beginning of the manuscript has clear rationale (Figures 1-3), but the story becomes convoluted in the latter parts (Figures 4-10). The authors should streamline the text to justify jumping between epithelial cells and macrophages, two different in vivo models, and explanations of why genes of interest in the expression data are important.

Reviewer #2: 1. The authors indicate that the IFA was identified as part of a larger natural compound screen, yet no data or citation is provided for this screen and its conditions. This may be important context for the approach and valuable information for the field.

2. The data concerning autophagy markers is quite confusing. Some markers in the IFA treated group are equal to or lower than that of the untreated control, whereas others are significantly increased compared to untreated and treated controls. Without a more thorough exploration of autophagy, it is difficult to draw high-confidence conclusions from these data, especially compared to the clarity of the other TLR4-mediates phenotypes.

3. Many of the citations in the manuscript are inappropriate or unclear. The authors should review their references and ensure they are citing relevant literature. Examples include:

a. Lines 75-78 - the referenced citation is not appropriate for this statement. Mike et al. systematically demonstrates linkages between capsule and hypermucoviscosity. This statement should be supported by primary literature or an appropriate primary literature review.

b. Lines 107-109 - the referenced citation is not appropriate for this statement. Ernst et al. specifically interrogates ST258 Kp, rather than hvKp.

c. Lines 111-114 - statements need citations.

d. Line 269 - citation 29 is not related to autophagy.

e. Line 281 - this review does not mention Kp.

f. Line 319, 332 - these are single studies; however, the authors' statement reflects the state of the field. This is not an appropriate way to support these statements.

Reviewer #3: This is an extremely large collection of experiments that I found somewhat counterproductive. One broad suggestion is to choose 2-4 key experiments per figure and either delete or move the rest to supplemental data. Figs 1 and 2 would be much more digestible if ~4 experiments were chosen and the rest removed due to their redundancy.

Many conclusions are overstated. For example, from Fig 1, the authors conclude that the reduction in CPS following IFA treatment is due to inhibition of CPS synthesis. It could be synthesis due to reduced enzymatic activity OR changes in cps or rmp gene expression. Alternatively, it could be that export is affected. From Fig 2, they state that HMV is reduced due to hindered CPS synthesis. While this could be true, they have not drawn a direct connection. There are reports of Klebsiella strains that retain the HMV phenotype even though CPS is reduced. Downregulation of the rmp genes could also lead to this phenotype. I have similar concerns over the frequent statements that IFA reduced capsule chain length based on Fig 1D (see details in comment below).

Nearly every experimental outcome is related to the reduced CPS/loss of HMV but the distinction, if any, between these two is not clear and I think is an important point to consider. Presumably this strain contains the rmpADC genes since it is HMV positive? Addition of a rmpD mutant could help elucidate whether the IFA-mediated virulence phenotypes are due to loss of HMV, reduced capsule, or potentially both (a rmpD mutant would be predicted to be HMV negative but have normal capsule production). This mutant would not need to be tested in all assays, but perhaps in some tissue culture assays and one of the in vivo models (mouse recommended, but I appreciate the time & expense may be prohibitive).

Please carefully review literature and ensure the appropriate references are cited. I believe a few statements were mis-referenced and at least one typo was noted.

This manuscript could benefit from some English language editing. There are a few non-sensical sentences and others with poorly-chosen or inappropriate descriptions that make it challenging to know what the authors are trying to communicate.

I believe it is PLoS policy that all data need to be accessible? I think this would mean the RNA-seq data presented in Fig 3 should be publicly available, and available to the reviewers.

**Part III – Minor Issues: Editorial and Data Presentation Modifications**

Reviewer #1: Minor Concerns:

1. Please clarify the phrasing of “TLR4-govern bacterial clearance-promoting immune responses”, specifically stated in Lines 54-55 but also mentioned similarly throughout the manuscript. Do the authors mean to say “TLR4-independent clearance of bacteria”? The current phrasing is confusing.

2. The sentence in Lines 75-78 is not supported by the indicated reference. Please update to the correct citation.

3. In the introduction, the authors emphasize the importance of CR-hvKp. Is the K7 strain used in this paper part a CR-hvKp strain? If so, this should be stated in the methods. If not, the introduction should be expanded to the importance of hvKp in general so as not to make the results seem relevant only to CR strains.

4. It would be interesting to include background on the drug screen that led to the identification of IFA. Perhaps a Supplementary figure could detail the design and top hits. This could also be a good resource for noting additional compounds that influence capsule inhibition.

5. Figure 1 uses multiple methodologies to indicate that IFA represses capsule. Of these approaches, the staining in 1E is the most subtle. I would suggest removing this panel as the data is much clearer in 1C and 1F.

6. In Figure 2C, how were the centrifugation values assessed? In Line 178 the authors describe “approximately one-third”, but there is no quantitative measure in the figure. If the data is qualitative only, then the quantitative language should be removed.

7. In Figure 4, the immunofluorescence is quite difficult to see, especially for the red channels. Perhaps a software like ImageJ could be used to quantitatively measure these signals. The authors should certainly include a secondary-only antibody control to confirm that the straining in question is not non-specific binding of the secondary antibody.

8. It has been previously noted that acapsular strains have similar intracellular survival within macrophages compared to WT strains (PMID 26045209). The authors should note this in the discussion as it may influence their findings surrounding TLR4 pathways.

9. The phrasing “IFA impeded the suppressive effect” in Lines 296-297 should be clarified as the double negative is confusing.

10. In Line 319, the word “reliable” is not quite correct. Galleria models are appropriate because they are accessible and feasible, but since these animals lack innate and adaptive responses of mammals, they are not necessarily used due their reliability.

11. The Galleria model should be explained for basic readout. For example, in Line 332 what does it mean for there to be an increase in melanization? How does this influence the readout of pathogenicity?

12. In Line 338, the word “established” in incorrect. Murine pneumonia models have been extensively and widely used throughout the field to study K. pneumoniae lung infections.

13. In Figure 8F, each panel should be labeled. It is unclear which groups are being referred to.

Reviewer #2: 1. Lines 45-46 are phrased awkwardly.

2. Lines 54, 253, - replace "-govern" with "-mediated."

3. Lines 66, 401 - omit "great epidemic." Kp infection does not meet the classical definition of an epidemic.

4. Lines 203-204 - these statements are not supported by image quantification or statistical analysis.

5. Line 248 - "significantly" indicates image quantification or statistical analysis, but none is shown.

6. Line 255 - this sentence needs revision.

7. Line 263 - "MPM" is not defined.

8. Line 268 - typo in TLR.

9. Lines 290-295 - I do not think that the authors have provided sufficient evidence to support their conclusion. It may be that the ratio of different PAMPs (LPS, capsule, etc.) modulate the interaction with TLR4, rather than it being specific to capsule recognition of TLR4.

10. Line 391 - most tlr4-/- mice appear to be dead at 32 hours in Figure 10A.

11. Line 391/3 - IFA enhances, rather than inhibits, expression of these genes.

12. Lines 391-392 - there does appear to be IFA-mediated enhancement of Mx1 expression in Figure 10D. Please clarify.

13. Lines 485-486 - which drugs are the authors referring to?

14. Line 510 - please provide methodological details for the capsule mutant or the relevant citation where it was created.

15. Lines 551-552 - where are MIC values reported?

16. All figures, please show exact p-values.

17. For figure 1C, 1H-I, 2A-B, 2E, 3F-G, 4F-H, 5A-J, 6, 7C-D, 8G-J, 9 A-E, 9F, 9G, and 10B-K, please show individual datapoints.

18. Figure 2A - the x-axis label is cut off.

19. Figure 2B - the y-axis is mislabelled.

20. Figure 2C - the statements in the text require quantification to be supported.

21. Figure 3C lacks clear labeling. Presumably "C" and "T" refer to control and treatment?

22. Figure 4A-B - the axis labels are condensed to the axis.

23. Figure 5K-M - it is unclear which groups are being compared.

24. Figure 8F - please label each histology image.

25. Figure 9F-K - please clarify the y-axis. Do the authors mean % of total cells for each marker set?

26. All figure legends - the figure legends indicate a comparison to vehicle control, but there are comparison bars on many figure panels. Please clarify.

27. Table 1 does not need to be in the main body.

Reviewer #3: Fig 1C, this is a relatively small decrease in UA. What amount of UA is detected in a capsule mutant? Knowing that baseline UA level would help interpret the significance of the decrease from IFA treatment.

Fig 1D, I agree with the authors that there is a decrease in the amount of high molecular weight capsule in this gel. However, it does not show the shift to lower molecular weight that occurs with loss of HMV as reported by Walker, et al. 2023 and Khadka, et al. 2023. With out that observation, it cannot necessarily be concluded that IFA reduces chain length. (example of an overstated conclusion mentioned above)

Fig. 1E, these images are not very effective. It is almost impossible to see the bacteria unless zoomed way in and even harder to see the CPS staining. I do not think this data adds to the already convincing data that IFA reduces CPS.

Fig. 1G, it is unclear how the CPS thickness would be measured between 100-200 nm based on the image provided, where it looks to be maybe 50 nm. Can the authors clarify this discrepancy? I recognize this is a representative image, but if the data are going to be quantified and graphed, the representative image should match the data in the graph.

Fig 2A, what is the rationale for two different centrifugal forces? No mention was made regarding these two forces, and I personally think one is sufficient to illustrate the HMV results. However, if both are to be shown, the Y-axes should be the same on these two graphs to better illustrate the differences.

Fig 2B, Y-axis label is incorrect

Fig 2C, it is not clear what the authors are attempting to demonstrate with this experiment. The statement in the results (lines 181-182) does not articulate a conclusion. It also seems somewhat redundant to the data shown in 2A,B. Recommend removing this unless there is new information that can be obtained from this.

Fig 2E, is redundant to 2A and adds no new information. What is the purpose of fold change in string test length? FC relative to what? This assay is highly subjective and should not be quantified. The sedimentation assay (as in 2A,B) is more quantifiable. Given the overwhelming amount of data in this paper, I recommend removing this (or just stating in Results).

Fig. 2F, it appears that for both K7 and ∆GT, the colonies on plates with IFA are smaller, suggesting a growth defect. Is this the same medium as used in Fig. 1B? This potential discrepancy warrants explaining.

Lines 201-203 the wrong reference is given, and I believe the authors are mis-stating some findings.

Ref 22 has author list/title that do not go together.

Fig 3 in general I found to be a bit of a distraction. Beyond demonstrating that the correlation between ATP levels and HMV status occurs in this strain (which is interesting!), the rest of this data was not particularly informative without further follow-up. At a minimum, I think it worth commenting whether expression of capsule synthesis genes or the rmp genes were impacted by IFA.

The title to Fig 3 is misleading. There is no data in this figure that relate to chain length, only the global impact of IFA on Klebsiella transcription.

Fig 4A,B, all other figures so far use 32 µg/ml. Why 64 here?

Fig 4C,D, what is the purpose of the lower set of images (presumably bright field, but this is not mentioned in the legend)? Nothing can be seen in the panel C T1 images.

Fig 4 title is mis-leading. Capsule is not eliminated by IFA

Fig 5A-J, please provide the gene name in a prominent place (such as with panel letter, or upper left corner of graph). These fonts are so small they can only be read when zoomed in.

Fig 5K-M, please keep Y-axes the same in all graphs for better comparison. I would also remove the 8µg/ml treatment as it does not add anything.

Fig 7C are K7 and K7+IFA mis-labeled?

Would be helpful if the color scheme in Fig 7 matched Fig 8&9

Fig 8F, please label each part with the strain used. There are 4 strains/conditions but 5 images. Presumably one is uninfected?

Fig 9F-K, please label the individual flow cytometry panels with the strain/condition.

PLOS authors have the option to publish the peer review history of their article (what does this mean?). If published, this will include your full peer review and any attached files.

Reviewer #1: No

Reviewer #2: **Yes: **Jay Vornhagen

Reviewer #3: No
---

## [Decision Letter · Decision Letter 1]

3 Nov 2024

PPATHOGENS-D-24-01091R1Isoferulic acid facilitates effective clearance of hypervirulent Klebsiella pneumoniae through targeting capsulePLOS Pathogens

Dear Dr. Deng,

Thank you for submitting your revised manuscript to PLOS Pathogens. After careful consideration, we feel that it has merit but does not fully meet PLOS Pathogens's publication criteria as it currently stands. As you will notice, all three reviewers agree that the revised manuscript is much more readable and substantially improved from the first submission. However, I will draw your attention to concerns raised by Reviewer 1 regarding different concentrations of DMSO used in IFA versus what was used in vehicle-only control (10% versus 1%). As mentioned by Reviewer 1, DMSO can independently modulate the host immune response. In the manuscript, IFA was dissolved in 10% DMSO, 45% Saline, 40% PEG400, and 5% Tween. Both the IFA treatment group and the control group must be treated with the correct vehicle-only control. Additionally, the solvent contents of the vehicle-only control must be mentioned in the manuscript. Therefore, we invite you to submit a revised version of the manuscript that addresses the points raised during the review process.

Please submit your revised manuscript within 60 days Jan 02 2025 11:59PM. If you will need more time than this to complete your revisions, please reply to this message or contact the journal office at plospathogens@plos.org. Please include the following items when submitting your revised manuscript:

We look forward to receiving your revised manuscript.

Kind regards,

M Ammar Zafar

Guest Editor

PLOS Pathogens

D. Scott Samuels

Section Editor

PLOS Pathogens

Michael Malim

Editor-in-Chief

PLOS Pathogens

orcid.org/0000-0002-7699-2064

 **Reviewers' Comments:**Reviewer's Responses to Questions

**Part I - Summary**

Reviewer #1: In this revised manuscript by Wang, et al., the authors have substantially improved their original article and clearly modified the manuscript to address many reviewer concerns. The study is certainly more streamlined in the revision by removing redundant experiments and tangential data.

While this manuscript has been substantially improved, I remain skeptical about the off-target effects of the vehicle control. Further, I remain unconvinced that the authors have uncovered novel biology in Figures 4-8 as these experiments largely confirm previous findings reported by other groups when studying the effects of acapsular strains on host response.

Reviewer #2: This manuscript is a resubmission by Wang, Yang et al. reports the findings of a study exploring the effects of isoferulic acid (IFA) on Klebsiella pneumoniae (Kp) capsule expression and resulting phenotypes. The authors have addressed many of the reviewer requests, and the data demonstrate a high degree of rigor. Unfortunately, in the the revision process, the most novel data (autophagy and TLR4) was removed. By removing these data, this study shows that IFA represses capsule, which in turn, reduces virulence. While these data are highly convincing, the relationship between capsule, mucoviscosity, and virulence are very well documented. Thus, in its current form this manuscript suffers from a lack impact.

Reviewer #3: The revised manuscript by Wang, et al. (PPATHOGENS-D-24-01091_R1) is much improved. The removal of several redundant experiments has made this a more readable paper. The conclusions are more appropriate for the data presented, and the inclusion of the data with the rmpD mutant (Fig S3) and the transcript data (Fig 3F) clarify that IFA more specifically impacts capsule rather than HMV. I think most of the reviewer comments have been sufficiently addressed and I have just a few remaining questions and comments.

**Part II – Major Issues: Key Experiments Required for Acceptance**

Reviewer #1: 1. In the response to reviewers, the authors state that the IFA vehicle is 10% DMSO, but that the experiments were carried out with vehicle "reduced DMSO to 1% and the PBS control was also treated with IFA vehicle". The vehicle control should be exactly what the vehicle is delivered in for the experimental condition. It is not appropriate to administer IFA with 10% DMSO but use 1% DMSO as the vehicle control. This is a log less of DMSO, a chemical that can influence the host. The authors should add the corrected DMSO control and clearly state that this control is used in each panel where the comparison is made.

2. The majority of the experiments in this manuscript remain detailing the host responses to acapsular Kp and not in understanding specifically how IFA is decreasing capsule. I wish that the drug screen identifying IFA was in larger focus and that more space was dedicated to understanding properties of drugs that inhibit capsule synthesis and not on reconfirming previous findings.

Reviewer #2: (No Response)

Reviewer #3: 1. Fig 1D is redundant, not terribly very convincing, and the quantification data are difficult to fully evaluate without a normalization step. Presumably equal ODs were loaded, but there is insufficient information about this. M&M state “equal amounts” but is this equal OD or equal culture volume?? If the authors wish to keep this in, I would remove the quantification. The uronic acid assay is sufficient to demonstrate a reduction in CPS.

2. Lines 160-161, the study referenced (Mike et al. 2021, ref 21) does not conclude that excessive CPS is required for HMV, just that some threshold level of CPS is required. This distinction was first reported by Walker, et al. 2020 (ref 36). It is true that some low CPS-producing mutants also lose the HMV phenotype, but it is important to distinguish that excess CPS and HMV can be distinct phenotypes. There may be some strains where it is hard to distinguish reduced CPS from HMV, and it is likely that the decrease in CPS caused by IFA in strain K7 drops below the threshold needed for the strain to become HMV positive. I would encourage cautious wording so as to not overgeneralize the findings.

3. RcsB and RfaH are major positive regulators of cps gene expression. That transcripts for both of these genes are significantly reduced suggests that IFA may be acting at a level upstream of cps gene expression. This information should be addressed in the Discussion.

4. Fig 5A and S3E, adhesion data I think is more typically presented as a percent of the inoculum. If the inocula values are available, I recommend reanalyzing the data. It is curious to me that in 5A the deltaGT mutant has much lower adherence than K7+IFA and I wonder if this is reflective of differences in the inoculum?

**Part III – Minor Issues: Editorial and Data Presentation Modifications**

Reviewer #1: (No Response)

Reviewer #2: 1. Please indicate which datapoint in S1B is IFA.

2. Line 199 "gud" is a typo.

3. Line 282 "huge" is not a common scientific term.

Reviewer #3: 1. Line 97, I’m not sure this is the appropriate reference for the statement made. I would guess the information might be included in that paper, but a more appropriate reference would be one of the more recent reviews on CPS synthesis and export.

2. Line 183, this section on transcriptomic changes due to IFA does not really address mechanism, as stated. It certainly provides insight, and narrows the target to cps gene expression and expression of cps regulators, but this is not “mechanism”. Please edit accordingly.

3. Line 215, the statement “while the amount of wild-type K7 began to decrease until 120 min” is somewhat confusing considering it decreases further at each timepoint, including 180 min. Please edit for clarity and accuracy.

4. Fig 5, why are the MOI different for epithelial cells vs macrophages? I realize this will not alter interpretation, but the rationale should be stated.

5. Please indicate the limit of detection and define the black lines (means, medians?) for the bacterial burden data in Galleria and mice (figs S3, 6, 7).

A few typos were noted:

Line 169, reference to colonies on LB agar, but figure shows blood agar

Line 199, replace gud with gnd

Line 249, something is not correct here. Perhaps the use of “liberating” is not appropriate?

Line 282, 321, 327, others?, folds should be fold (not plural).

PLOS authors have the option to publish the peer review history of their article (what does this mean?). If published, this will include your full peer review and any attached files.

Reviewer #1: No

Reviewer #2: **Yes: **Jay Vornhagen

Reviewer #3: No

**Figure resubmission:**
---

## [Editor Report · Decision Letter 2]

27 Nov 2024

Dear Dr. Deng,

We are pleased to inform you that your manuscript 'Isoferulic acid facilitates effective clearance of hypervirulent Klebsiella pneumoniae through targeting capsule' has been provisionally accepted for publication in PLOS Pathogens. 

Best regards,

M Ammar Zafar

Guest Editor

PLOS Pathogens

D. Scott Samuels

Section Editor

PLOS Pathogens

Michael Malim

Editor-in-Chief

PLOS Pathogens

orcid.org/0000-0002-7699-2064
---

## [Editor Report · Acceptance letter]

9 Dec 2024

Dear Dr. deng,

We are delighted to inform you that your manuscript, "Isoferulic acid facilitates effective clearance of hypervirulent Klebsiella pneumoniae through targeting capsule," has been formally accepted for publication in PLOS Pathogens.

Best regards,

Sumita Bhaduri-McIntosh

Editor-in-Chief

PLOS Pathogens

orcid.org/0000-0003-2946-9497

Michael Malim

Editor-in-Chief

PLOS Pathogens

orcid.org/0000-0002-7699-2064